# RESPONSE MODELING OF HYPER-PARAMETERS FOR DEEP CONVOLUTIONAL NEURAL NETWORKS

## ABSTRACT

Hyper-parameter optimization (HPO) is critical in training high performing Deep Neural Networks (DNN). Current methodologies fail to define an analytical *response surface* (Bergstra & Bengio, 2012) and remain a training bottleneck due to their use of additional internal hyper-parameters and lengthy manual evaluation cycles. We demonstrate that the low-rank factorization of the convolution weights of intermediate layers of a CNN can define an analytical response surface. We quantify how this surface acts as an auxiliary to optimizing training metrics. We introduce a fully autonomous dynamic tracking algorithm – autoHyper – that performs HPO on the order of hours for various datasets including ImageNet and requires no manual intervention or a priori knowledge. Our method – using a single RTX2080Ti – is able to select a learning rate within 59 hours for AdaM (Kingma & Ba, 2014) on ResNet34 applied to ImageNet and improves in top-1 test accuracy by **4.93%** over the default learning rate. In contrast to previous methods, we empirically prove that our algorithm and response surface generalize well across model, optimizer, and dataset selection removing the need for extensive domain knowledge to achieve high levels of performance.

## 1 INTRODUCTION

The choice of Hyper-Parameters (HP) – such as initial learning rate, batch size, and weight decay – has shown to greatly impact the generalization performance of Deep Neural Network (DNN) training (Keskar et al., 2017; Wilson et al., 2017; Li et al., 2019; Yu & Zhu, 2020). By increasing the complexity of network architectures (from high to low parameterized models) and training datasets (class number and samples), the manual intervention to tune these parameters for optimization becomes a practically expensive and highly challenging task. Therefore, the problem of Hyper-Parameter Optimization (HPO) becomes central to developing highly efficient training workflows.

Recent studies shift the gear toward development of a meaningful *metric measure* to explain effective HP tuning for DNN training. This is done in several behavioural studies, including changes in loss surfaces (Keskar et al., 2017), input perturbation analysis (Novak et al., 2018), and the energy norm of the covariance of gradients (Jastrzebski et al., 2020), just to name a few. In fact, the abstract formulation of the HPO problem, as highlighted by Bergstra & Bengio (2012), can be modelled by

$$\lambda^* \leftarrow \arg\min_{\lambda \in \Lambda} \{ \mathbb{E}_{x \sim M} [\mathcal{L}(x; \mathcal{A}_\lambda(X^{(\text{train})})] \}, \tag{1}$$

where, $X^{(\text{train})}$ and $x$ are random variables, modelled by some natural distribution $M$, that represent the train and validation data, respectively, $\mathcal{L}(\cdot)$ is some expected loss, and $\mathcal{A}_\lambda(X^{(\text{train})})$ is a learning algorithm that maps $X^{(\text{train})}$ to some learned function, conditioned on the hyper-parameter set $\lambda$. Note that this learned function, denoted as $f(\theta; \lambda; X^{(\text{train})})$, involves its own inner optimization problem. The HPO in (1) highlights two optimization problems of which optimization over $\lambda$ cannot occur until optimization over $f(\theta; \lambda; X^{(\text{train})})$ is complete. This fact applies heavy computational burden for HPO. Bergstra & Bengio (2012) reduce this burden by attempting to solve the following

$$\lambda^* \leftarrow \arg\min_{\lambda \in \Lambda} \tau(\lambda), \tag{2}$$

where $\tau$ is called the *hyper-parameter response function* or *response surface*, and $\Lambda$ is some set of choices for $\lambda$ (*i.e.* the search space). The goal of the response surface is to introduce an auxiliary

function parameterized by $\lambda$ of which its minimization is directly correlated to minimization of the objective function $f(\theta)$. Little advancements in an analytical model of the response surface has led to estimating it by (a) running multiple trials of different HP configurations (*e.g.* grid searching), using evaluation against validation sets as an estimate to $\tau$; or (b) characterizing the distribution model of a configuration's performance metric (*e.g.* cross-validation performances) to numerically define a relationship between $\tau$ and $\lambda$.

An important shift occurred when Bergstra & Bengio (2012) showed that random searching is more efficient to grid searching, particularly when optimizing high-dimensional HP sets. To mitigate the time complexity and increase overall performance, subsequent methods attempted to characterize the distribution model for such random configurations (Snoek et al., 2012; Eggensperger et al., 2013; Feurer et al., 2015a;b; Klein et al., 2017; Falkner et al., 2018) or employed population control (Young et al., 2015; Jaderberg et al., 2017) or early-stopping (Karnin et al., 2013; Li et al., 2017; 2018). However, these methods suffer from (a) additional internal HPs that require manual tuning facilitated by extensive domain knowledge; (b) heavy computational overhead whereby the optimization process takes days to weeks in most cases (Li et al., 2017; Falkner et al., 2018; Yu & Zhu, 2020); (c) poor generalization across model selection, datasets, and general experimental configurations (*e.g.* optimizers); and (d) strong dependence on a manually defined search ranges that heavily influences results (Choi et al., 2020; Sivaprasad et al., 2020). Importantly, these ranges are generally chosen based on intuition, expert domain knowledge, or some form of a priori knowledge.

In this paper, we employ the notion of knowledge gain (Hosseini & Plataniotis, 2020) to model a response surface – solvable with low computational overhead – and use it to perform automatic HPO that does not require any a priori knowledge while still achieving competitive performance against baselines and existing state of the art (SOTA) methods. Our goal is therefore to develop an algorithm that is fully autonomous and domain independent that can achieve competitive performance (not necessarily superior performance). We restrict our response surface to consider a single HP, namely the initial learning rate $\eta$, and support this choice by noting that the initial learning rate is the most sensitive and important HP towards final model performance (Goodfellow et al., 2016; Bergstra & Bengio, 2012; Yu & Zhu, 2020) (see also Figure 10 in Appendix C). We demonstrate how our method's optimization directly correlates to optimizing model performance. Finally, we provide empirical measures of the computational requirements of our algorithm and present thorough experiments on a diverse set of Convolutional Neural Network (CNN) and Computer Vision dataset that demonstrate the generalization of our response surface.

The main contributions of this work are as follows:

1. Inspired by knowledge gain, we introduce a well-defined, analytical response surface using the low-rank-factorization of convolution weights (Equation 5).

2. We propose a dynamic tracking algorithm of low computational overhead on the order of minutes and hours, dubbed autoHyper, to optimize our response surface and conduct HPO.

3. This algorithm requires no domain knowledge, human intuition, or manual intervention, and is not bound by a manually set searching space, allowing for completely automatic setting of the initial learning rate; a novelty for deep learning practitioners.

## 1.1 RELATED WORKS

We leave extensive analysis of the related works to established surveys (Luo, 2016; He et al., 2019; Yu & Zhu, 2020) but present a general overview here. Grid searching and manual tuning techniques that require extensive domain knowledge trial various configurations and retain the best. Random search (Bergstra & Bengio, 2012) was proven to be more efficient, particularly in high-dimensional cases, but these methods suffer from redundancy and high computational overhead. Bayesian optimization (Snoek et al., 2012; Eggensperger et al., 2013; Feurer et al., 2015a;b; Klein et al., 2017) techniques attempt to characterize the distribution model of the random HP configurations. They fail to properly define the response surface $\tau$ and resolve to estimating it by rationalizing a Gaussian process over sampling points. The use of neural networks over Gaussian to model the generalization performance was shown to have better computational performance (Snoek et al., 2015; Springenberg et al., 2016). Furthermore, the early stopping methods (Karnin et al., 2013; Li et al., 2017; 2018) spawn various configurations with equal resource distributions, successively stopping poor-performing configurations and reassigning resources dynamically. Population-based training (PBT)

methods (Young et al., 2015; Jaderberg et al., 2017) follow an evolutionary approach by spawning various experimental configurations and adapting poor-performing trials to warm restart with inherited learnable parameters and HPs. In addition, other methods such as orthogonal array tuning (Zhang et al., 2019), box-constrained derivative-free optimization (Diaz et al., 2017), reverse dynamics algorithm for SGD optimization (Maclaurin et al., 2015), and hybrid methods (Swersky et al., 2013; 2014; Domhan et al., 2015; Falkner et al., 2018; Kandasamy et al., 2016) exist but demonstrate no significant benefits over the previous techniques. Generally, each of these methods suffer from high computational overheads – on the order of days to weeks to converge – as well as additional internal HPs that heavily influence performance and generalization. In recent years, many Python libraries have also been developed that include these optimization methods (Bergstra et al., 2013; Kotthoff et al., 2017; Akiba et al., 2019).

## 2 A New Response Surface Model

In this section, we motivate and develop a new response surface model $\tau(\lambda)$ based on the low-rank factorization of convolutional weights in a CNN. Unlike the common approach of cross-validation performance measures, we define a new measure on the well-posedness of the intermediate layers of a CNN and relate this measure to the general performance of the network. We first start by adopting the low-rank measure of convolution weights.

### 2.1 Knowledge Gain via Low-Rank Factorization

Consider a four-way array (4-D tensor) $\mathbf{W} \in \mathbb{R}^{N_1 \times N_2 \times N_3 \times N_4}$ as the convolution weights of an intermediate layer of a CNN ($N_1$ and $N_2$ being the height and width of kernel size, and $N_3$ and $N_4$ to the input and output channel size, respectively). Under the convolution operation, the input feature maps $\boldsymbol{F}^I \in \mathbb{R}^{W \times H \times N_3}$ are mapped to an arbitrary output feature map $\boldsymbol{F}^O \in \mathbb{R}^{W \times H \times N_4}$ by

$$\boldsymbol{F}^O_{:,:,i_4} = \sum_{i_3=1}^{N_3} \boldsymbol{F}^I_{:,:,i_3} * \mathbf{W}_{:,:,i_3,i_4}.$$

$$\mathbf{W} \text{ (4-D Tensor)} \xrightarrow{\text{unfold}} \boldsymbol{W}_d \text{ (2-D Matrix)} \xrightarrow{\text{factorize then decompose}} \underbrace{\widehat{\boldsymbol{U}}_d \widehat{\Sigma}_d \widehat{\boldsymbol{V}}_d^T}_{\widehat{\boldsymbol{W}}_d} + \boldsymbol{E}_d$$

We note the importance of factorizing the unfolded matrix $\boldsymbol{W}_d$ using a low-rank factorization (we use the Variational Bayesian Matrix Factorization (VBMF) (Nakajima et al., 2013)). Without this factorization, the presence of noise will inhibit proper analysis. This noise $\boldsymbol{E}_d$ will "capture" the randomness of initialization and ignoring it will allow us to better analyze our unfolded matrices and make our response surface robust to initialization method.

Following the definition of *Knowledge Gain (KG)* from Hosseini & Plataniotis (2020), one can now define a metric for each network layer using the norm energy of the low-rank factorization as

$$\mathcal{G}_d(\widehat{\boldsymbol{W}}_d) = \frac{1}{N_d \cdot \sigma_1(\widehat{\boldsymbol{W}}_d)} \sum_{i=1}^{N_d'} \sigma_i(\widehat{\boldsymbol{W}}_d). \tag{3}$$

where, $\sigma_1 \geq \sigma_2 \geq \ldots \geq \sigma_{N_d}$ are the associated low-rank singular values in descending order. Here $N_d = \mathrm{rank}\{\widehat{\boldsymbol{W}}_d\}$ and the unfolding can be done in either input or output channels i.e. $d \in \{3, 4\}$. For more information on KG as well as its efficient algorithmic computation, we refer the reader to Hosseini & Plataniotis (2020).

The metric defined in (3) is normalized such that $\mathcal{G}_d \in [0, 1]$ and can be used to probe CNN layers to monitor their efficiency in the carriage of information from input to output feature maps. We can further parameterize the KG by the HP set $\lambda$, epoch $t$, and network layer $\ell$ as $\bar{\mathcal{G}}_{d,t,\ell}(\lambda)$. A perfect network and set of HPs would yield $\bar{\mathcal{G}}_{d,T,\ell}(\lambda) = 1 \quad \forall \ell \in [L]$ where $L$ is the number of layers in the network and $T$ is the last epoch. In this case, network layer functions as a better autoencoder through iterative training and the carriage of information throughout the network is maximized. Conversely, $\bar{\mathcal{G}}_{d,T,\ell}(\lambda) = 0$ indicates that the information flow is very weak such that the mapping is effectively random ($\| \boldsymbol{E}_d \|$ is maximized.).

## 2.2 DEFINITION OF NEW RESPONSE FUNCTION

Interestingly, if $\bar{\mathcal{G}}_{d,t,\ell}(\lambda) = 0$ in early stages of training, it is evidence that no learning has occurred, indicative of an initial learning rate that is too small (no progress has been made to reduce the randomization). It then becomes useful to track the zero-valued KGs within a network's intermediate layers' input and output channels, which effectively becomes a measure of channel rank. We denote this rank per epoch as follows:

$$\mathcal{Z}_t(\lambda) \leftarrow \frac{1}{2L} \sum_{\ell \in [L]} \sum_{d \in \{3,4\}} \lfloor 1 - \bar{\mathcal{G}}_{d,t,\ell}(\lambda) \rfloor$$

where $\mathcal{Z}_t(\lambda) \in [0, 1)$. Finally, we define the average rank across $T$ epochs as

$$\mathcal{Z}(\lambda) \leftarrow \frac{1}{T} \sum_{t \in [T]} \mathcal{Z}_t(\lambda). \tag{4}$$

Note that $\mathcal{Z}(\lambda) \in [0, 1)$. The average rank measure in (4) is therefore a normalized summation of the zero-valued singular values of the low-rank factorization across all layers' input and output unfolded tensor arrays.

Relating to the notion of HPO and the response surface, we return to (1) and (2). Where previously the nature of these two optimization problems was poorly understood or practically unsolvable, we propose a new problem that is well understood and practically solvable (on the computational order of hours). To solve for the optimal HP set $\lambda$, we look at the following optimization problem

$$\lambda^* \leftarrow \arg\min_{\lambda} 1 - \mathcal{Z}(\lambda), \text{ subject to } \| \nabla_\lambda \mathcal{Z}(\lambda) \|_2^2 \leq \epsilon \tag{5}$$

where $\epsilon \in [0, 1)$ is some small conditioning error. Returning to equation 2, our response surface is therefore defined as $\tau = 1 - \mathcal{Z}(\lambda)$ subject to $\| \nabla_\lambda \mathcal{Z}(\lambda) \|_2^2 \leq \epsilon$. Note that we now simplify our problem to only consider $\lambda = \eta$. Also, we do not explicitly calculate the gradient $\nabla_\lambda \mathcal{Z}(\lambda)$, but rather use this constraint to guide our dynamic tracking algorithm (see section 3). To explain this somewhat counterintuitive formulation, we analyze Figures 1(a) & 2, which demonstrate that as learning rates increase, $\mathcal{Z}(\eta)$ plateaus to zero. Specifically, we notice that optimal learning rates lie towards the inception of the plateau of $\mathcal{Z}(\eta)$, before $\mathcal{Z}(\eta) = 0$. This can also be seen in Figures 8 & 7 in Appendix A. Therefore, we wish to design our response surface such that the solution lies at the inception of the plateau-ing region (see the red dot in Figure 1(a)). In this case, our constraint $\| \nabla_\lambda \mathcal{Z}(\lambda) \|_2^2 \leq \epsilon$ promotes learning rates that lie along this plateau-ing region, while $\arg\min_{\lambda} 1 - \mathcal{Z}(\lambda)$ promotes, of those learning rates in the plateau-ing region, a learning rate that lies towards the inception of this plateau-ing region.

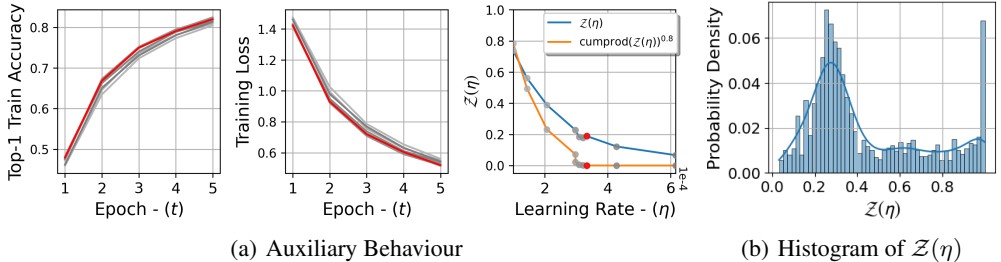

(a) Auxiliary Behaviour    (b) Histogram of $\mathcal{Z}(\eta)$

Figure 1: (a) Auxiliary representation of $\mathcal{Z}(\eta)$ to training loss and training accuracy. The red dot indicates the learning rate chosen by our method with corresponding metrics drawn in red lines. Lines and scatter points in grayscale show various trialled learning rates. (b) Distribution of $\mathcal{Z}(\eta)$ taken from the searching phase of autoHyper over numerous experimental configurations applied on CIFAR10, CIFAR100, TinyImageNet, and ImageNet (see subsection 4.1).

More generally, high $\mathcal{Z}(\eta)$ indicates a learning rate that is too small, as intermediate layers do not make sufficient progress in early stages of learning and therefore their KGs remain very low. This observation follows that of Li et al. (2019) in which larger initial learning rates result in better

generalization performance. Promotion of these larger learning rates is therefore achieved by our gradient constraint in Equation 5. Conversely, too large of a learning rate can over-regulate a network and, therefore, we wish not to minimize $\mathcal{Z}(\eta)$ completely but tune it to be sufficiently small to arrive at the inception of its plateau, creating a sort of kick-start if you will. This is achieved by the balance between our minimization and constraint in Equation 5.

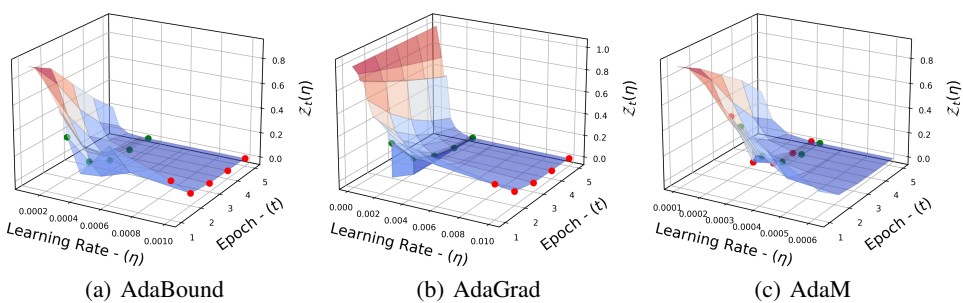

(a) AdaBound            (b) AdaGrad            (c) AdaM

Figure 2: $\mathcal{Z}(\eta)$ for various learning rates using AdaBound, AdaGrad, and AdAM on ResNet34 applied to CIFAR10. The author-suggested initial learning rate is indicated by the red markers, and the autoHyper suggested learning rate is indicated by the green markers.

Finally, we choose $T = 5$ in our experiment. The early phase of training has been shown to be an important criterion in optimal model performance (Jastrzebski et al., 2020; Li et al., 2019) and we therefore wish to only consider 5-epochs to ensure optimization within this phase. Additionally, Figure 2 and Figures 8 & 7 in Appendix A tell us that $\mathcal{Z}(\eta)$ stabilizes after 5 epochs.

## 2.3 Empirical Evaluation of New Response Model

Figure 1(a) visualizes results of our method on ResNet34 on CIFAR10 optimized with AdaM. The learning rate selected by our method results in lowest training loss and highest top-1 training accuracy over the 5-epoch range we consider. We note the importance of stopping at the inception of this plateau region as even though higher learning rates, highlighted by the gray-scale lines/markers, result in potentially lower $\mathcal{Z}(\eta)$, they do not guarantee lower training losses or higher training accuracies. We conclude that our response surface is a strong auxiliary to training loss, and optimizing HPs relative to our response surface will in fact optimize towards training loss and accuracy.

Figure 1(b) displays the histogram of $\mathcal{Z}(\eta)$ values over various experimental configurations (see subsection 4.1). Note the presence of a multimodal distribution that peaks at low $\mathcal{Z}(\eta)$ but importantly not zero. This visualizes our method's tendency to converge to a consistent range of values for irrespective of experimental configuration, showing the generalization of our response surface.

## 3 AutoHyper: Automatic Tuning of Initial Learning Rate

The pseudo-code for autoHyper is presented in Algorithm 3. Analyzing equation 5, we state that the optimal solution lies within the inception of the plateauing region of $\mathcal{Z}(\eta)$. To find this region, autoHyper first initializes a logarithmic grid space, from $\eta_{min} = 1 \times 10^{-4}$ to $\eta_{max} = 0.1$ of $S = 20$ step sizes, denoted by $\mathbf{\Omega}$. It iterates through each $\eta_i \in \mathbf{\Omega}; i \in \{0, \ldots, 19\}$, and computes $\mathcal{Z}(\eta)$, until a plateau is reached. Once a plateau is reached, $\mathbf{\Omega}$ is reset such that $\eta_{min}$ and $\eta_{max}$ "zoom" towards the learning rates at the plateau. This process is repeated recursively until no significant difference between $\eta_{min}$ and $\eta_{max}$ exists. On average, this recursion occurs 3 to 4 times and as shown in Figure 3, the number of trialled learning rates remains very low (between 10-30 on average). Importantly, our algorithm is not constrained by its initial grid space. As it tracks $\mathcal{Z}(\eta)$ over learning rates, it may grow and shrink its search bounds dynamically. This permits our method to be fully autonomous and require no human intuition in setting of the initial grid space.

---

**Algorithm 1** autoHyper

---

**Require:** grid space function $\Psi$, learning rate significant difference delta $\alpha = 5 \times 10^{-5}$, and rate of change function $\zeta$

1: **procedure** RESTART( )
2:     learning rate index $i = 0$
3:     $\Omega = \Psi(\eta_{min}, \eta_{max}, S)$
4: **end procedure**
5: RESTART( )
6: **while** True **do**
7:     **if** $i = |\Omega|$ **then**    // increase search space since no plateau has been found yet
8:         set $\eta_{min} = \eta_{max}$, increase $\eta_{max}$ and RESTART( )
9:     **end if**
10:     **if** $\eta_{max} - \eta_{min} < \alpha$ **then**    // limits of search space are not significantly different
11:         **return** $\eta_{max}$
12:     **end if**
13:     with $\eta_i \leftarrow \Omega_i$, train for 5 epochs
14:     compute rank: $\mathcal{Z}(\eta_i)$ per equation 4
15:     **if** $\mathcal{Z}(\eta_i) = 1.0$ **then**    // all KG is zero-valued, $\eta_{min}$ is too small
16:         increase $\eta_{min}$ and RESTART( )
17:     **end if**
18:     **if** $i = 0$ and $\mathcal{Z}(\eta_i) < 0.5$ and this is the first run **then**    // initial $\eta_{min}$ is too large
19:         reduce $\eta_{min}$ and RESTART( )
20:     **else**
21:         **if** $\mathcal{Z}(\eta_i) = 0.0$ **then**    // all KG is non-zero, don't search further, perform "zoom"
22:             set $\eta_{min} = \Omega_{i-2}, \eta_{max} = \Omega_i$ and RESTART( )
23:         **end if**
24:         compute rate of change of $\mathcal{Z}(\eta_i)$: $\delta \leftarrow \zeta(\{\mathcal{Z}(\eta_0), \dots, \mathcal{Z}(\eta_i)\})$
25:         **if** rate of change plateaus **then**    // perform "zoom"
26:             set $\eta_{min} = \Omega_{i-1}, \eta_{max} = \Omega_i$ and RESTART( )
27:         **end if**
28:     **end if**
29:     $i \mathrel{+}= 1$
30: **end while**

---

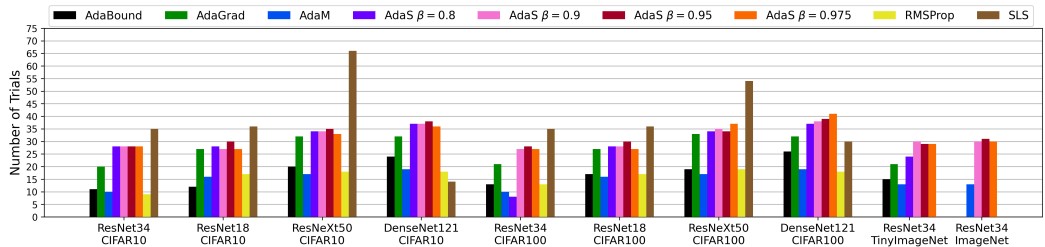

Figure 3: Computational analysis of autoHyper over various setups (number of learning rates autoHyper trialled before converging). ResNet34 trials take 3 minutes, 3 minutes, 18 minutes, and 220 minutes for CIFAR10, CIFAR100, TinyImageNet and, ImageNet, respectively. ResNet18, ResNeXt50, and DenseNet121 trials take 2 minutes, 3 minutes, and 3 minutes respectively for both CIFAR10 and CIFAR100.

We note here that the choice of $\Psi$ and $\zeta$ mentioned in Algorithm 3 (*i.e.* grid space and rate of change functions, respectively) will have a significant affect on the final generated learning rate. We make use of numpy's *geomspace* function for the logarithmic grid spacing, and calculate the rate of change in $\mathcal{Z}(\eta)$ by taking the cumulative product of the sequence of $\mathcal{Z}(\eta_i)$, to the power of $0.8$. A logarithmic grid space is used as our response surface is more sensitive to smaller learning rates (see Figure 2). Note that initial grid bounds are not important as our algorithm can shift those bounds dynamically, however the successive increments between learning rates in the grid must be sufficiently small (on the order of $1 \times 10^{-4}$ as in our initialization). Since our response surface itself is not guaranteed to monotonically decrease, as shown in Figure 4(b), we employ the cumulative product of $\mathcal{Z}(\eta_i)$ (as our rate of change function), which is a monotonically decreasing function

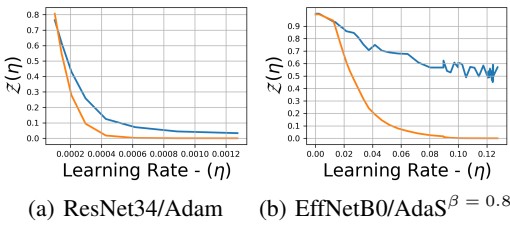

(a) ResNet34/Adam    (b) EffNetB0/AdaS$^{\beta=0.8}$

Figure 4: $\mathcal{Z}(\eta)$ (blue) vs. cumprod($\mathcal{Z}(\eta)$)$^{0.8}$ (orange) for (a) a stable and (b) an unstable architecture on CIFAR10.

– since $\mathcal{Z}(\eta_i) \in [0,1)$ – and is therefore always guaranteed to converge. The cumulative product (to the power of $0.8$) is a good choice because it (a) is always guaranteed to plateau (since $0 \le \mathcal{Z}(\eta_i) < 1$), which removes the need for some manually tuned threshold and (b) because it dampens noise well. Because the cumulative product on its own degrades to zero rather quickly in many scenarios, raising it to the power of $0.8$ regulates this effect. This power is technically tune-able, however we show empirically in Figure 4(a) and 4(b) that $0.8$ behaves well for both stable and unstable architectures. Refer to Figure 9 in Appendix C for the performance results of EfficientNetB0.

## 4 EXPERIMENTS

In this section, we conduct an ablative study of our algorithm autoHyper and response surface on various network architectures trained using various optimizers and applied to image classification datasets. We also compare autoHyper against existing SOTA; Random Search.

### 4.1 EXPERIMENTAL SETUPS

**Ablative study.** All experiments are run using an RTX2080Ti, 3 cores of an Intel Xeon Gold 6246 processor, and 64 gigabytes of RAM. In our ablative study, we run experiments on CIFAR10 (Krizhevsky et al., 2009), CIFAR100 (Krizhevsky et al., 2009), TinyImageNet (Li et al.), and ImageNet (Russakovsky et al., 2015). On CIFAR10 and CIFAR100, we apply ResNet18 (He et al., 2015), ResNet34 (He et al., 2015), ResNeXt50 (Xie et al., 2016), and DenseNet121 (Huang et al., 2017). On TinyImageNet and ImageNet, we apply ResNet34. For architectures applied to CIFAR10 and CIFAR100, we train using AdaM (Kingma & Ba, 2014), AdaBound (Luo et al., 2019), AdaGrad (Duchi et al., 2011), RMSProp (Tieleman & Hinton, 2012), AdaS$^{(\beta = \{0.8, 0.9, 0.95, 0.975\})}$ (Hosseini & Plataniotis, 2020) (with early-stop), and SLS (Vaswani et al., 2019). For ResNet34 applied to TinyImageNet, we train using AdaM, AdaBound, AdaGrad, and AdaS$^{(\beta = \{0.8, 0.9, 0.95, 0.975\})}$. For ResNet34 applied to ImageNet, we train using AdaM and AdaS$^{(\beta = \{0.9, 0.95, 0.975\})}$. Note that the $\beta$ in AdaS-variants is known as the *gain factor* and trades between performance and convergence rate : a low $\beta$ converges faster but at the cost of performance and vice-versa. For each experimental setup we ran one training sequence using suggested learning rates (baseline) and one training sequence using learning rates generated by autoHyper (see Tables 1-4 in Appendix B). Refer to Appendix B for additional details on the ablative study.

**Comparison to Random Search.** Because Random Search generally requires iterative manual refinement and is highly sensitive to the manually set search space (Choi et al., 2020; Sivaprasad et al., 2020), we attempt a fair comparison by providing the same initial search space that autoHyper starts with, and allow for the same number of trials that autoHyper takes (see Figure 3). We note however that this does provide the Random Search with a slight advantage since a priori knowledge of how many trials to consider is not provided to autoHyper. See Appendix D for additional detail.

### 4.2 RESULTS

**Consistent performance across architectures, datasets, and optimizers.** We visualize the primary results of each experiment in Figure 5(a) (additional results are shown in Figure 11 in Appendix C). From these figures we see how well our method generalizes to experimental configurations by noting the consistency in top-1 test accuracies when training using the autoHyper generated initial learning rate vs. the baseline. Further, we note that if there is loss of performance when using an initial learning rate generated by autoHyper, we identify that this loss is $< 1\%$ in all experiments except three: On CIFAR100, the baselines of ResNeXt50 trained using AdaM, ResNext50 trained using RMSProp, and DenseNet121 trained using AdaBound achieve $1.2\%$, $2.28\%$ and $1.9\%$ better top-1 test accuracy, respectively. We note importantly however that when accounting for the standard deviation of each of these results, only the DenseNet121 experiement maintains its $> 1\%$ improvement. Refer to Appendix C (Tables 5-8) to see the tabulated results of each experiment. We also importantly highlight how autoHyper is able to generalize across experimental setup whereas Random Search cannot (see Figure 5(b)). Because Random Search (and other SOTA methods) depend heavily on their manually defined parameters such as epoch budget or initial search space,

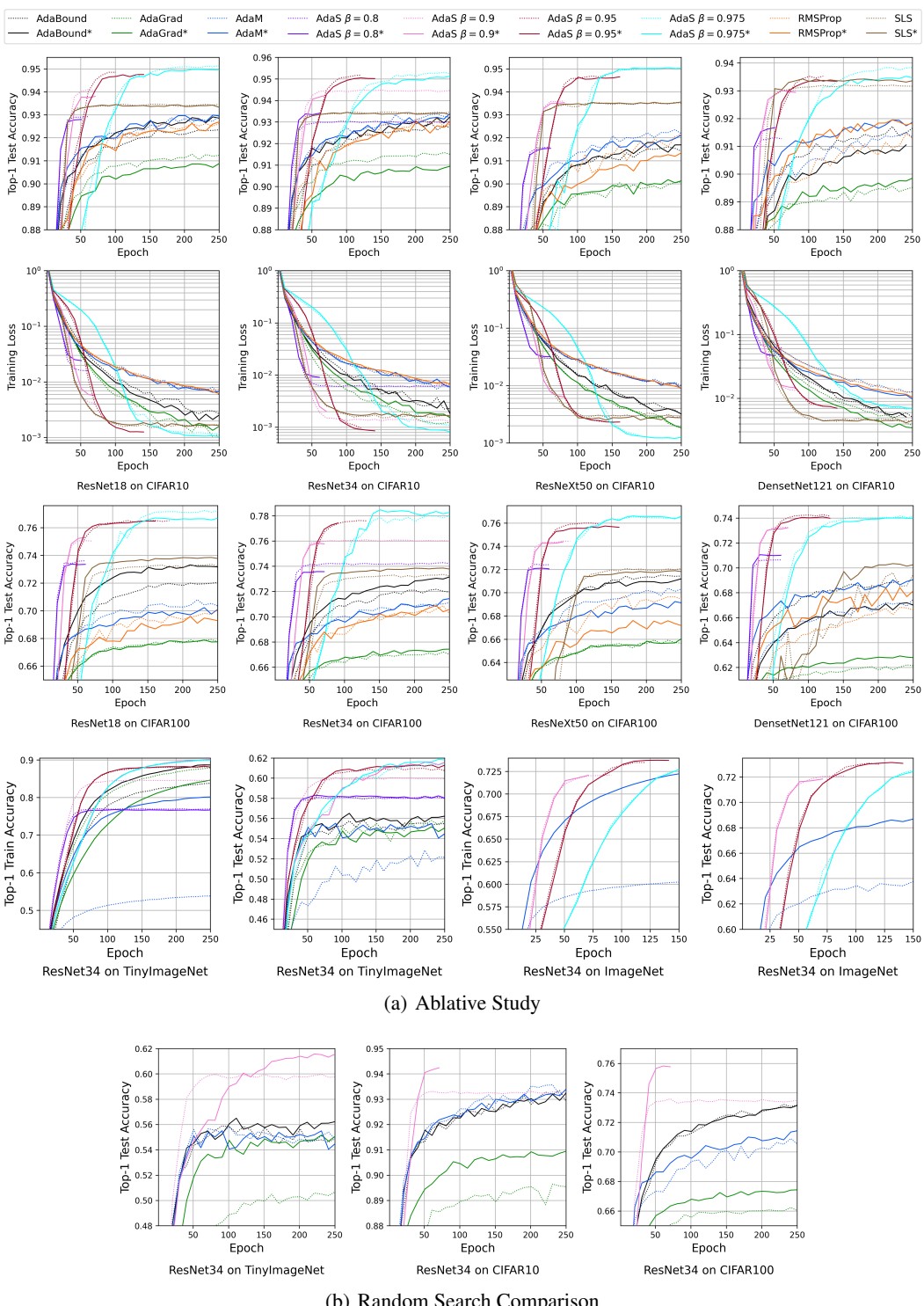

(a) Ablative Study

(b) Random Search Comparison

Figure 5: Results of the (a) ablative study and (b) Random Search comparison experiments. Titles below plots indicate what experiment the above plots refers to. Legend labels marked by '*' (solid lines) show results for autoHyper generated learning rates and dotted lines are the (a) baselines and (b) Random Search results.

generalization to experimental setup is not feasible, as demonstrated here. In contrast, we have shown that autoHyper is perfectly capable of performing well no matter the experimental setting without need for manual intervention/refinement of any kind; a novelty.

**Fully autonomous discovery of optimal learning rates.** Importantly, we highlight how our method is able to fully autonomously tune the initial learning and achieve very competitive performance. Whereas traditional HPO methods (like Random Search) are extremely sensitive to initialization of the search space, which would normally require extensive domain or a priori knowledge to set, our method is not: given a new dataset, model, and/or other hyper-parameter configurations, a practitioner could use simply call our algorithm to automatically set a very competitive initial learning rate. If truly superior performance is required, one could perform more extensive HPO around the autoHypersuggested learning rate, removing the need to perform iterative manual refinement.

**Superior performance over existing SOTA.** As visualized in Figure 13, although Random Search proves competitive for Adabound and AdaM applied on CIFAR10 and CIFAR100, it cannot find a competitive learning rate for $AdaS^{\beta = 0.9}$ or AdaGrad and performs worse for AdaM applied on TinyImageNet. AdaGrad applied on TinyImageNet loses as much as 4% top-1 test accuracy. This highlights how autoHyper can automatically find more competitive learning rates to a Random Search given the same computational budget, and with significantly less manual intervention. These results additionally highlight why validation loss (or accuracy) cannot be used as a substitute to our metric (see Figure 14 in subsection D.2 for additional discussion).

**Drastic improvements in AdaM applied to TinyImageNet and ImageNet.** ResNet34 trained using AdaM and applied to TinyImageNet and ImageNet achieves final improvements of 3.14% and 4.93% in top-1 test accuracy, respectively (see Table 5 in Appendix C). Such improvements come at a minimal cost using our method, requiring 13 trials (4 hours) and 16 trials (59 hours) for TinyImageNet and ImageNet, respectively (see Figure 3).

**Extremely fast and consistent convergence rates.** We visualize the convergence rates of our method in Figure 3. Importantly, we identify the consistency of required trials per optimizer across architecture and dataset selection as well as the low convergence times. We identify that the longest convergence time for our method is on ResNet34 trained using $AdaS^{\beta = 0.95}$ applied to ImageNet, which took 31 trials and a total of 114 hours. We note that our method exhibits less consistent results when optimizing using SLS as SLS tends to result in high $\mathcal{Z}(\eta)$ over multiple epochs and different learning rates. Despite this, our model still converges and results in competitive performance.

**Performance improvement over increased epoch budgets.** In reference to Table 6 in Appendix C, we highlight how, of the 29 experimental configurations, when trained using the initial learning rate suggested by autoHyper, only 12 of them outperform the baseline. However, as training progresses, we note that by the end of the fixed epoch budget, 18 of the 29 experiments trained using the initial learning rate suggested by autoHyper outperform the baselines. Further, in many of the cases where baselines perform better, they remain within the standard deviation of trials, and are therefore not significantly better. These results are surprising as our goal with this method was to achieve competitive results in tuning the initial learning rate however, in more than half the cases, our method results in increased performance at a significantly smaller computational cost.

## 5 CONCLUSION

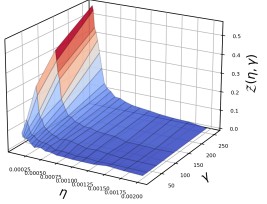

Figure 6: $\mathcal{Z}(\eta, \gamma)$ for learning rate ($\eta$) and mini-batch size ($\gamma$).

In this work we proposed an analytical response surface that acts as auxiliary to training metrics and generalizes well. We proposed an algorithm, autoHyper, that solves this surface and quickly generates learning rates that are competitive to author-suggested and Random Search-suggested values – in some cases, even drastically superior. We therefore have introduced an algorithm that can perform HPO fully autonomously and extremely efficiently, and resolves many of the drawbacks of current SOTA. Figure 6 visualizes our response surface over a multi-dimension HP set and highlights how our response surface remains solvable. We identify that autoHyper could be adapted to simultaneously optimize multiple HPs by tracking tangents across this surface towards the minimum, but leave this to future work.

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

## A    RANK BEHAVIOUR OVER MULTIPLE EPOCHS

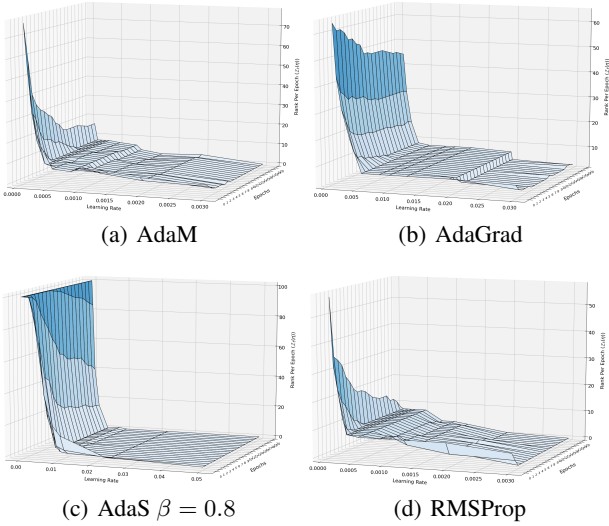

(a) AdaM

(b) AdaGrad

(c) AdaS $\beta = 0.8$

(d) RMSProp

Figure 7: Rank ($\mathcal{Z}(\eta)$) for various learning rates on VGG16 trained using AdaM, AdaGrad, AdaS $\beta = 0.8$, and RMSProp and applied to CIFAR10. A fixed epoch budget of 20 was used. We highlight how across these 20 epochs, very little progress is made beyond the first first epochs. It is from this analysis that we choose our epoch range of $T = 5$.

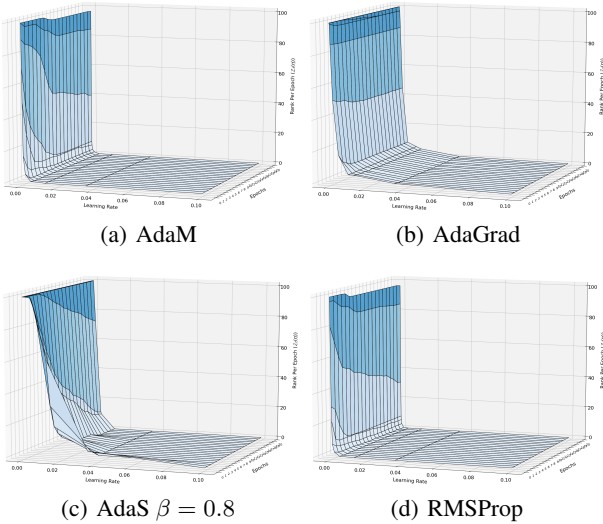

(a) AdaM

(b) AdaGrad

(c) AdaS $\beta = 0.8$

(d) RMSProp

Figure 8: ($\mathcal{Z}(\eta)$) for various learning rates on ResNet34 trained using AdaM, AdaGrad, AdaS $\beta = 0.8$, and RMSProp and applied to CIFAR10. A fixed epoch budget of 20 was used. We highlight how across these 20 epochs, very little progress is made beyond the first first epochs. It is from this analysis that we choose our epoch range of $T = 5$.

## B    ADDITIONAL EXPERIMENTAL DETAILS FOR SUBSECTION 4.1

We note the additional configurations for our experimental setups.

**Datasets:** For CIFAR10 and CIFAR100, we perform random cropping to $32 \times 32$ and random horizontal flipping on the training images and make no alterations to the test set. For TinyImageNet,

we perform random resized cropping to $64 \times 64$ and random horizontal flipping on the training images and center crop resizing to $64 \times 64$ on the test set. For ImageNet, we follow He et al. (2015) and perform random resized cropping to $224 \times 244$ and random horizontal flipping and $256 \times 256$ resizing with $224 \times 224$ center cropping on the test set.

**Additional Configurations:** Experiments on CIFAR10, CIFAR100, and TinyImageNet used mini-batch sizes of 128 and ImageNet experiments used mini-batch sizes of 256. For weight decay, $5 \times 10^{-4}$ was used for AdaS-variants on CIFAR10 and CIFAR100 experiments and $1 \times 10^{-4}$ for all optimizers on TinyImageNet and ImageNet experiments, with the exception of AdaM using a weight decay of $7.8125 \times 10^{-6}$. For AdaS-variant, the momentum rate for momentum-SGD was set to 0.9. All other hyper-parameters for each respective optimizer remained default as reported in their original papers. For CIFAR10 and CIFAR100, we use the manually tuned suggested learning rates as reported in Wilson et al. (2017) for AdaM, RMSProp, and AdaGrad. For TinyImageNet and ImageNet, we use the suggested learning rates as reported in each optimizer's respective paper. Refer to Tables 1-4 to see exactly which learning rates were used, as well as the learning rates generated by autoHyper. CIFAR10, CIFAR100, and TinyImageNet experiments were trained for 5 trials with a maximum of 250 epochs and ImageNet experiments were trained for 3 trials with a maximum of 150 epochs. Due to AdaS' stable test accuracy behaviour as demonstrated by Hosseini & Plataniotis (2020), an early-stop criteria, monitoring testing accuracy, was used for CIFAR10, CIFAR100, and ImageNet experiments. For CIFAR10 and CIFAR100, a threshold of $1 \times 10^{-3}$ for AdaS$^{\beta = 0.8}$ and $1 \times 10^{-4}$ for AdaS$^{\beta = \{0.9, 0.95\}}$ and patience window of 10 epochs. For ImageNet, a threshold of $1 \times 10^{-4}$ for AdaS$^{\beta = \{0.8, 0.9, 0.95\}}$ and patience window of 20 epochs. No early stop is used for AdaS$^{\beta = 0.975}$.

**Learning Rates:** We report every learning rate in Tables 1-4.

Table 1: Learning rates for ResNet34 experiments. Left inner columns show suggested, right inner columns show autoHyper generated. Note that the superscript for AdaS-variants indicates their $\beta$ gain factor.

| Optimizer | CIFAR10 | | CIFAR100 | | TinyImageNet | | ImagetNet | |
|---|---|---|---|---|---|---|---|---|
| AdaM | 0.0003 | 0.000333 | 0.0003 | 0.000241 | 0.001 | 0.0001965 | 0.001 | 0.0001965 |
| AdaBound | 0.001 | 0.000347 | 0.001 | 0.000347 | 0.001 | 0.0000944 | - | - |
| AdaGrad | 0.01 | 0.002861 | 0.01 | 0.002236 | 0.01 | 0.0022359 | - | - |
| AdaS$^{(0.8)}$ | 0.03 | 0.012374 | 0.03 | 0.010401 | 0.03 | 0.010185 | - | - |
| AdaS$^{(0.9)}$ | 0.03 | 0.012374 | 0.03 | 0.010190 | 0.03 | 0.0085857 | 0.02 | 0.011479 |
| AdaS$^{(0.95)}$ | 0.03 | 0.015336 | 0.03 | 0.015025 | 0.03 | 0.0085857 | 0.02 | 0.011479 |
| AdaS$^{(0.975)}$ | 0.03 | 0.012374 | 0.03 | 0.010190 | 0.03 | 0.0085857 | 0.02 | 0.011479 |
| RMSProp | 0.0003 | 0.000168 | 0.0003 | 0.000197 | - | - | - | - |
| SLS | 1.0 | 0.034191 | 1.0 | 0.034191 | - | - | - | - |

Table 2: Learning rates for ResNet18 experiments. Left inner columns show suggested, right inner columns show autoHyper generated.

| Optimizer | CIFAR10 | | CIFAR100 | |
|---|---|---|---|---|
| AdaM | 0.0003 | 0.0006756 | 0.0003 | 0.0006756 |
| AdaBound | 0.001 | 0.00036040 | 0.001 | 0.00024896 |
| AdaGrad | 0.01 | 0.0049724 | 0.01 | 0.0049724 |
| AdaS$^{(\beta = 0.8)}$ | 0.03 | 0.0126594 | 0.03 | 0.0126594 |
| AdaS$^{(\beta = 0.9)}$ | 0.03 | 0.0104254 | 0.03 | 0.0126594 |
| AdaS$^{(\beta = 0.95)}$ | 0.03 | 0.01042544 | 0.03 | 0.01042543 |
| AdaS$^{(\beta = 0.975)}$ | 0.03 | 0.0104254 | 0.03 | 0.007071 |
| RMSProp | 0.0003 | 0.0004697 | 0.0003 | 0.0004697 |
| SLS | 1.0 | 0.03419134 | 1.0 | 0.0341913 |

Table 3: Learning rates for ResNeXt50 experiments. Left inner columns show suggested, right inner columns show autoHyper generated.

| Optimizer | CIFAR10 | | CIFAR100 | |
|---|---|---|---|---|
| AdaM | 0.0003 | 0.0006756 | 0.0003 | 0.0006756 |
| AdaBound | 0.001 | 0.00036040 | 0.001 | 0.00024896 |
| AdaGrad | 0.01 | 0.0049724 | 0.01 | 0.0049724 |
| AdaS$^{(\beta = 0.8)}$ | 0.03 | 0.0126594 | 0.03 | 0.0126594 |
| AdaS$^{(\beta = 0.9)}$ | 0.03 | 0.0104254 | 0.03 | 0.0126594 |
| AdaS$^{(\beta = 0.95)}$ | 0.03 | 0.023134 | 0.03 | 0.022666 |
| AdaS$^{(\beta = 0.975)}$ | 0.03 | 0.0104254 | 0.03 | 0.007071 |
| RMSProp | 0.0003 | 0.0004697 | 0.0003 | 0.0004697 |
| SLS | 1.0 | 0.03419134 | 1.0 | 0.0341913 |

Table 4: Learning rates for DenseNet121 experiments. Left inner columns show suggested, right inner columns show autoHyper generated.

| Optimizer | CIFAR10 | | CIFAR100 | |
|---|---|---|---|---|
| AdaM | 0.0003 | 0.0020109 | 0.0003 | 0.00093162 |
| AdaBound | 0.001 | 0.0030176 | 0.001 | 0.003147979 |
| AdaGrad | 0.01 | 0.01537206 | 0.01 | 0.01537206 |
| AdaS$^{(\beta = 0.8)}$ | 0.03 | 0.06107176 | 0.03 | 0.03978104 |
| AdaS$^{(\beta = 0.9)}$ | 0.03 | 0.0598363 | 0.03 | 0.050414 |
| AdaS$^{(\beta = 0.95)}$ | 0.03 | 0.0492772 | 0.03 | 0.0357048479 |
| AdaS$^{(\beta = 0.975)}$ | 0.03 | 0.0598363 | 0.03 | 0.0504143 |
| RMSProp | 0.0003 | 0.00201093 | 0.0003 | 0.00067564 |
| SLS | 1.0 | 0.0031184 | 1.0 | 0.087943 |

## C  ADDITIONAL RESULTS FOR SUBSECTION 4.2

**Large deviation from the suggested initial learning rates.** Referring to Tables 1-4 & 9, we notice variation in autoHyper suggested learning rates as compared to the author-suggested and Random Search-selected ones. The learning rates generated by our method reveal the "blind spots" that the authors originally overlooked in their HPO. Interestingly, however, we note the similarity in initial learning for ResNet34 trained using AdaM on CIFAR10, and can confirm this as an optimal learning rate. Importantly, our method is significantly quicker than the grid searching technique employed by Wilson et al. (2017). **Observations on the generalization characteristics of optimizers.** Figure 5(a) identifies the poor generalization characteristics of AdaM, AdaBound, AdaGrad, and SLS where they consistently achieve low training losses, but do not exhibit equivalently high top-1 test accuracies. We note that these observations are similar to those made by Wilson et al. (2017); Li et al. (2019); Jastrzebski et al. (2020). We additionally contribute that AdaS does generalize well. We also highlight SLS' multi-order-of-magnitude tolerance to initial learning rate as well as the stability of the AdaS optimizer, particularly when applied on TinyImageNet.

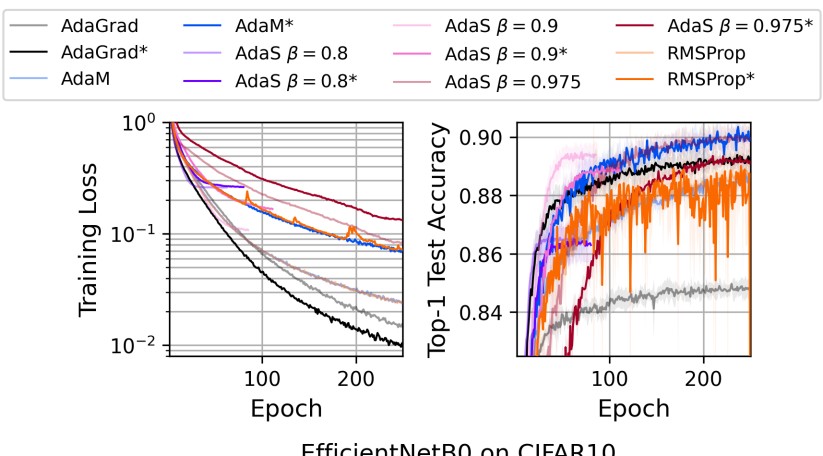

Figure 9: Test accuracy and trianing loss for EfficientNetB0 applied to CIFAR100. Importantly, EfficientNetB0 is an unstable network architecture in relation to our response surface and yet our method, autoHyper, is still able to converge and achieve competitive performance.

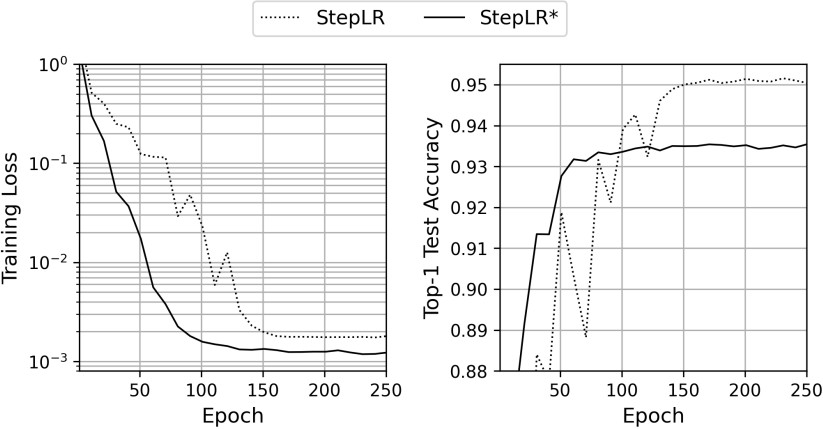

Figure 10: Demonstration of the importance of initial learning rate in scheduled learning rate case, for ResNet18 applied on CIFAR10, using Step-Decay method with step-size = 25 epochs and decay rate = 0.5. As before, the dotted line represents the baseline results, with initial learning rate = 0.1, and the solid line represents the results using autoHyper's suggested learning rate of 0.008585. These results highlight the importance of initial learning rate, even when using a scheduled learning rate heuristic, and demonstrates the importance of the additional step-size and decay rate hyperparameters. Despite better initial performance from the autoHyper suggest learning rate, the step-size and decay rate choice cause the performance to plateau too early.

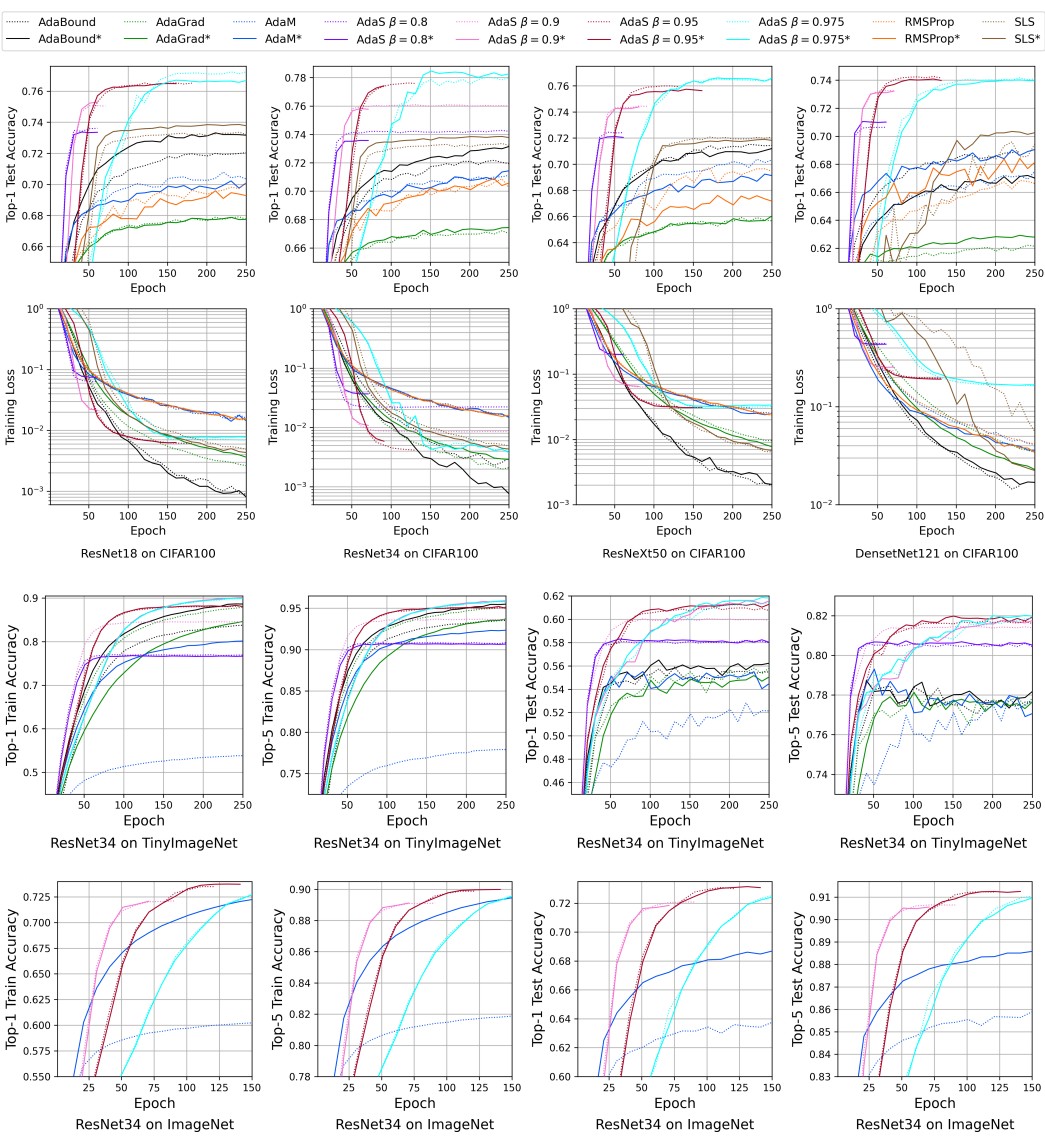

Figure 11: Full results of CIFAR100, TinyImageNet, and ImageNet experiments. Top-1 test accuracy and training losses are reported for CIFAR100 experiments and top-1 and top-5 test and training accuracies are reported for TinyImageNet and ImageNet. Titles below the figures indicate to which experiments the above figures belong to. As before, lines indicated by the '*' (solid lines), are results using initial learning rate as suggested by autoHyper.

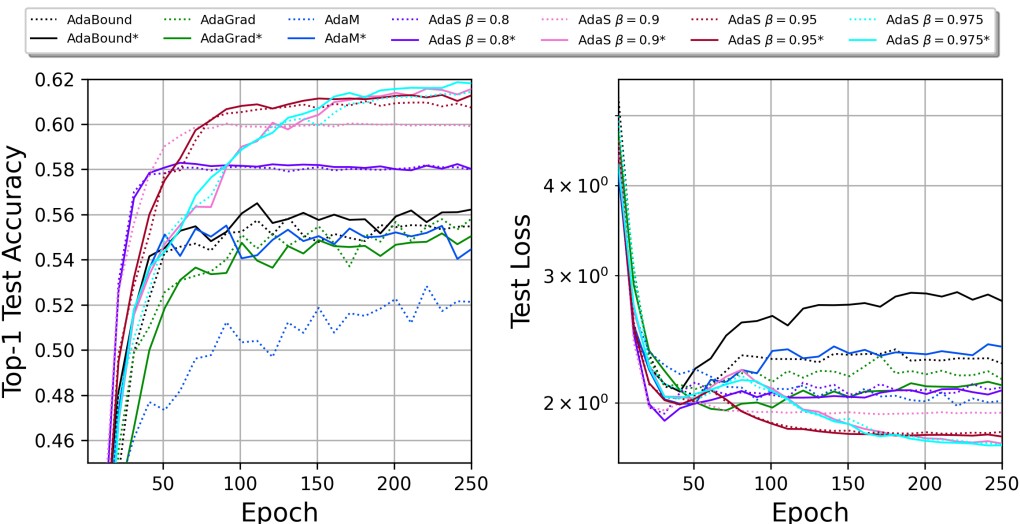

Figure 12: Top-1 Test Accuracy and Test Loss for ResNet34 Experiments applied on TinyImageNet. As before, lines indicated by the '*' (solid lines), are results using initial learning rate as suggested by autoHyper. These results visualize the inconsistency in tracking test loss as a metric to optimize final testing accuracy. This can be seen, for example, when looking at the test loss and test accuracy plots for AdaM, where the test loss for the baseline is lower than that of the autoHyper suggested results but autoHyper achieves better test accuracy. These results also highlight the instability of tracking testing accuracy or less instead of the metric defined in Equation 5.

Table 5: List of ResNet34 top-1 test accuracy using various optimizers and applied to various datasets. Note that each cell reports the average accuracy over each experimental trial, sub-scripted by the standard deviation over the trials. Values on the left are from trials trained using the initial learning rate as generated by autoHyper. Values on the right are from trials trained using the suggested initial learning rate. Note a '*' indicates that early-stop has been activated.

| Dataset | Optimizer | Epoch 50 | Epoch 100 | Epoch 150 | Epoch 200 | Epoch 250 |
|---|---|---|---|---|---|---|
| CIFAR10 | AdaBound | $\mathbf{91.66_{\pm0.33}}/90.84_{\pm0.17}$ | $\mathbf{92.37_{\pm0.25}}/92.03_{\pm0.44}$ | $\mathbf{92.78_{\pm0.20}}/92.67_{\pm0.14}$ | $\mathbf{92.79_{\pm0.42}}/92.59_{\pm0.13}$ | $\mathbf{93.24_{\pm0.17}}/92.74_{\pm0.16}$ |
| | AdaGrad | $89.41_{\pm0.39}/\mathbf{90.17_{\pm0.61}}$ | $90.30_{\pm0.33}/\mathbf{91.10_{\pm0.25}}$ | $90.77_{\pm0.35}/\mathbf{91.34_{\pm0.37}}$ | $90.67_{\pm0.30}/\mathbf{91.40_{\pm0.15}}$ | $90.94_{\pm0.14}/\mathbf{91.50_{\pm0.24}}$ |
| | AdaM | $91.44_{\pm0.19}/\mathbf{91.54_{\pm0.35}}$ | $\mathbf{92.72_{\pm0.15}}/92.71_{\pm0.27}$ | $92.93_{\pm0.21}/\mathbf{93.07_{\pm0.33}}$ | $93.06_{\pm0.34}/\mathbf{93.17_{\pm0.30}}$ | $\mathbf{93.39_{\pm0.31}}/93.22_{\pm0.36}$ |
| | AdaS$^{0.8}$ | $\mathbf{93.35_{\pm0.14}}/93.04_{\pm0.13}$ | $\mathbf{93.40^*_{\pm0.15}}/93.03_{\pm0.16}$ | $\mathbf{93.40^*_{\pm0.15}}/93.00_{\pm0.17}$ | $\mathbf{93.40^*_{\pm0.15}}/93.02_{\pm0.16}$ | $\mathbf{93.40^*_{\pm0.15}}/93.02_{\pm0.13}$ |
| | AdaS$^{0.9}$ | $94.00_{\pm0.13}/\mathbf{94.35_{\pm0.22}}$ | $94.25^*_{\pm0.22}/\mathbf{94.48_{\pm0.13}}$ | $94.25^*_{\pm0.22}/\mathbf{94.45_{\pm0.11}}$ | $94.25^*_{\pm0.22}/\mathbf{94.47_{\pm0.13}}$ | $94.2^*5_{\pm0.22}/\mathbf{94.46_{\pm0.12}}$ |
| | AdaS$^{0.95}$ | $91.90_{\pm0.25}/\mathbf{92.03_{\pm0.33}}$ | $94.91_{\pm0.16}/\mathbf{95.14_{\pm0.16}}$ | $95.08^*_{\pm0.18}/\mathbf{95.20_{\pm0.11}}$ | $95.08^*_{\pm0.18}/\mathbf{95.20_{\pm0.11}}$ | $95.08^*_{\pm0.18}/\mathbf{95.20_{\pm0.11}}$ |
| | AdaS$^{0.975}$ | $87.68_{\pm0.95}/\mathbf{88.91_{\pm1.30}}$ | $92.92_{\pm0.32}/\mathbf{93.09_{\pm0.52}}$ | $94.75_{\pm0.12}/\mathbf{94.99_{\pm0.23}}$ | $95.01_{\pm0.13}/\mathbf{95.14_{\pm0.34}}$ | $95.13_{\pm0.11}/\mathbf{95.24_{\pm0.15}}$ |
| | RMSProp | $90.69_{\pm0.47}/\mathbf{90.86_{\pm0.58}}$ | $91.41_{\pm0.55}/\mathbf{91.59_{\pm0.77}}$ | $92.22_{\pm0.68}/\mathbf{92.69_{\pm0.33}}$ | $\mathbf{92.94_{\pm0.33}}/92.88_{\pm0.30}$ | $\mathbf{93.03_{\pm0.23}}/92.90_{\pm0.29}$ |
| | SLS | $\mathbf{93.30_{\pm0.16}}/93.28_{\pm0.10}$ | $93.41_{\pm0.09}/\mathbf{93.48_{\pm0.09}}$ | $93.39_{\pm0.10}/\mathbf{93.49_{\pm0.09}}$ | $93.34_{\pm0.13}/\mathbf{93.41_{\pm0.08}}$ | $93.33_{\pm0.06}/\mathbf{93.45_{\pm0.16}}$ |
| CIFAR100 | AdaBound | $\mathbf{69.21_{\pm0.59}}/68.02_{\pm0.75}$ | $\mathbf{71.38_{\pm0.44}}/70.57_{\pm0.40}$ | $\mathbf{72.39_{\pm0.27}}/71.67_{\pm0.49}$ | $\mathbf{72.83_{\pm0.16}}/72.08_{\pm0.27}$ | $\mathbf{73.15_{\pm0.24}}/71.94_{\pm0.66}$ |
| | AdaGrad | $\mathbf{65.35_{\pm0.46}}/65.15_{\pm0.27}$ | $\mathbf{66.72_{\pm0.34}}/66.58_{\pm0.38}$ | $\mathbf{67.03_{\pm0.50}}/66.91_{\pm0.31}$ | $\mathbf{67.16_{\pm0.50}}/66.97_{\pm0.25}$ | $\mathbf{67.43_{\pm0.59}}/67.02_{\pm0.23}$ |
| | AdaM | $68.31_{\pm0.48}/\mathbf{68.66_{\pm0.46}}$ | $69.71_{\pm0.63}/\mathbf{69.78_{\pm0.27}}$ | $70.43_{\pm0.29}/\mathbf{70.45_{\pm0.42}}$ | $\mathbf{70.98_{\pm0.43}}/70.61_{\pm0.33}$ | $\mathbf{71.43_{\pm0.28}}/71.11_{\pm0.37}$ |
| | AdaS$^{0.8}$ | $73.58_{\pm0.31}/\mathbf{74.18_{\pm0.32}}$ | $73.58^*_{\pm0.36}/\mathbf{74.21_{\pm0.35}}$ | $73.58^*_{\pm0.36}/\mathbf{74.22_{\pm0.35}}$ | $73.58^*_{\pm0.36}/\mathbf{74.19_{\pm0.24}}$ | $73.58^*_{\pm0.36}/\mathbf{74.21_{\pm0.26}}$ |
| | AdaS$^{0.9}$ | $75.54_{\pm0.25}/\mathbf{75.64_{\pm0.25}}$ | $75.78^*_{\pm0.21}/\mathbf{76.02_{\pm0.10}}$ | $75.78^*_{\pm0.21}/\mathbf{76.00_{\pm0.13}}$ | $75.78^*_{\pm0.21}/\mathbf{76.05_{\pm0.11}}$ | $75.78^*_{\pm0.21}/\mathbf{75.99_{\pm0.09}}$ |
| | AdaS$^{0.95}$ | $71.25_{\pm0.90}/\mathbf{71.57_{\pm1.00}}$ | $77.48^*_{\pm0.37}/\mathbf{77.53_{\pm0.18}}$ | $77.48^*_{\pm0.37}/\mathbf{77.60_{\pm0.22}}$ | $77.48^*_{\pm0.37}/\mathbf{77.60_{\pm0.22}}$ | $77.48^*_{\pm0.37}/\mathbf{77.60_{\pm0.22}}$ |
| | AdaS$^{0.975}$ | $62.11_{\pm2.01}/\mathbf{62.49_{\pm1.60}}$ | $\mathbf{75.15_{\pm0.26}}/74.48_{\pm0.53}$ | $\mathbf{78.44_{\pm0.34}}/77.81_{\pm0.23}$ | $\mathbf{78.15_{\pm0.22}}/77.76_{\pm0.38}$ | $\mathbf{78.26_{\pm0.35}}/78.00_{\pm0.28}$ |
| | RMSProp | $67.17_{\pm1.00}/\mathbf{67.69_{\pm0.62}}$ | $68.78_{\pm1.29}/\mathbf{69.28_{\pm0.29}}$ | $\mathbf{70.23_{\pm0.50}}/69.96_{\pm0.48}$ | $70.20_{\pm0.49}/\mathbf{70.39_{\pm0.50}}$ | $\mathbf{70.57_{\pm0.40}}/70.25_{\pm0.29}$ |
| | SLS | $\mathbf{66.81_{\pm1.25}}/64.88_{\pm1.23}$ | $\mathbf{73.48_{\pm0.24}}/73.02_{\pm0.20}$ | $\mathbf{73.78_{\pm0.19}}/73.14_{\pm0.21}$ | $\mathbf{73.74_{\pm0.32}}/73.24_{\pm0.11}$ | $\mathbf{73.77_{\pm0.12}}/73.22_{\pm0.11}$ |
| TinyImageNet | AdaBound | $\mathbf{55.07_{\pm1.06}}/54.15_{\pm1.13}$ | $\mathbf{55.75_{\pm0.34}}/55.37_{\pm0.15}$ | $\mathbf{56.66_{\pm0.30}}/55.00_{\pm0.85}$ | $\mathbf{56.06_{\pm0.04}}/55.05_{\pm0.58}$ | $\mathbf{56.22_{\pm0.17}}/55.48_{\pm0.67}$ |
| | AdaGrad | $52.01_{\pm0.43}/\mathbf{52.75_{\pm0.60}}$ | $54.01_{\pm0.39}/\mathbf{54.27_{\pm0.77}}$ | $54.46_{\pm0.68}/\mathbf{54.94_{\pm0.60}}$ | $54.69_{\pm0.30}/\mathbf{55.12_{\pm0.27}}$ | $55.04_{\pm0.54}/\mathbf{55.81_{\pm0.84}}$ |
| | AdaM | $\mathbf{53.98_{\pm1.14}}/47.43_{\pm1.60}$ | $\mathbf{54.67_{\pm0.93}}/48.97_{\pm1.86}$ | $\mathbf{55.86_{\pm0.26}}/52.22_{\pm0.51}$ | $\mathbf{54.81_{\pm0.73}}/50.14_{\pm1.20}$ | $\mathbf{54.46_{\pm1.14}}/52.13_{\pm1.14}$ |
| | AdaS$^{0.8}$ | $\mathbf{58.06_{\pm0.54}}/57.85_{\pm0.55}$ | $\mathbf{58.20_{\pm0.42}}/58.06_{\pm0.48}$ | $\mathbf{58.13_{\pm0.34}}/57.99_{\pm0.36}$ | $\mathbf{58.18_{\pm0.43}}/58.16_{\pm0.40}$ | $\mathbf{58.02_{\pm0.42}}/57.98_{\pm0.44}$ |
| | AdaS$^{0.9}$ | $54.45_{\pm0.91}/\mathbf{58.95_{\pm0.45}}$ | $58.99_{\pm0.25}/\mathbf{59.97_{\pm0.33}}$ | $\mathbf{60.08_{\pm0.40}}/59.97_{\pm0.40}$ | $\mathbf{61.38_{\pm0.49}}/60.01_{\pm0.30}$ | $\mathbf{61.56_{\pm0.58}}/59.91_{\pm0.45}$ |
| | AdaS$^{0.95}$ | $56.18_{\pm1.36}/\mathbf{57.41_{\pm0.71}}$ | $\mathbf{60.83_{\pm0.10}}/60.29_{\pm0.30}$ | $\mathbf{61.17_{\pm0.28}}/60.72_{\pm0.06}$ | $\mathbf{61.10_{\pm0.38}}/61.03_{\pm0.14}$ | $\mathbf{61.28_{\pm0.44}}/60.74_{\pm0.20}$ |
| | AdaS$^{0.975}$ | $54.87_{\pm1.22}/\mathbf{55.19_{\pm0.74}}$ | $58.91_{\pm0.57}/\mathbf{59.19_{\pm0.49}}$ | $\mathbf{60.62_{\pm0.69}}/60.22_{\pm0.27}$ | $\mathbf{61.70_{\pm0.39}}/61.13_{\pm0.37}$ | $\mathbf{61.81_{\pm0.45}}/61.44_{\pm0.27}$ |
| ImageNet | AdaM | $\mathbf{66.22_{\pm0.08}}/62.14_{\pm0.29}$ | $\mathbf{68.17_{\pm0.07}}/63.17_{\pm0.21}$ | $\mathbf{68.68_{\pm0.24}}/63.75_{\pm0.08}$ | - | - |
| | AdaS$^{0.9}$ | $\mathbf{71.61_{\pm0.10}}/71.48_{\pm0.35}$ | $71.87^*_{\pm0.20}/\mathbf{72.11_{\pm0.16}}$ | $71.87^*_{\pm0.20}/\mathbf{72.11_{\pm0.16}}$ | - | - |
| | AdaS$^{0.95}$ | $67.52_{\pm0.47}/\mathbf{68.15_{\pm0.21}}$ | $\mathbf{72.88_{\pm0.03}}/72.75_{\pm0.34}$ | $\mathbf{73.09_{\pm0.23}}/73.05_{\pm0.17}$ | - | - |
| | AdaS$^{0.975}$ | $\mathbf{58.25_{\pm0.34}}/58.09_{\pm0.36}$ | $69.17_{\pm0.09}/\mathbf{69.42_{\pm0.24}}$ | $72.42_{\pm0.15}/\mathbf{72.52_{\pm0.03}}$ | - | - |

Table 6: List of ResNet18 top-1 test accuracies using various optimizers and applied to various datasets. Note that each cell reports the average accuracy over each experimental trial, sub-scripted by the standard deviation over the trials. Values on the left are from trials trained using the initial learning rate as generated by autoHyper. Values on the right are from trials trained using the suggested initial learning rate. Note a '*' indicates that early-stop has been activated.

| Dataset | Optimizer | Epoch 50 | Epoch 100 | Epoch 150 | Epoch 200 | Epoch 250 |
|---|---|---|---|---|---|---|
| CIFAR10 | AdaBound | $\mathbf{91.24_{\pm 0.21}}/90.35_{\pm 0.51}$ | $\mathbf{92.25_{\pm 0.21}}/91.64_{\pm 0.42}$ | $\mathbf{92.69_{\pm 0.14}}/92.05_{\pm 0.25}$ | $\mathbf{92.79_{\pm 0.24}}/92.20_{\pm 0.26}$ | $\mathbf{92.85_{\pm 0.06}}/92.35_{\pm 0.18}$ |
| | AdaGrad | $89.42_{\pm 0.22}/\mathbf{89.67_{\pm 0.40}}$ | $90.39_{\pm 0.18}/\mathbf{90.79_{\pm 0.30}}$ | $90.55_{\pm 0.14}/\mathbf{91.16_{\pm 0.23}}$ | $90.75_{\pm 0.13}/\mathbf{91.20_{\pm 0.26}}$ | $90.87_{\pm 0.14}/\mathbf{91.23_{\pm 0.25}}$ |
| | AdaM | $\mathbf{91.43_{\pm 0.42}}/91.29_{\pm 0.45}$ | $\mathbf{92.17_{\pm 0.36}}/92.16_{\pm 0.13}$ | $\mathbf{92.63_{\pm 0.15}}/92.37_{\pm 0.17}$ | $\mathbf{92.88_{\pm 0.32}}/92.65_{\pm 0.21}$ | $\mathbf{92.95_{\pm 0.24}}/92.93_{\pm 0.22}$ |
| | AdaS$^{0.8}$ | $92.81_{\pm 0.13}/\mathbf{92.87_{\pm 0.23}}$ | $92.80^*_{\pm 0.16}/\mathbf{92.92_{\pm 0.19}}$ | $92.80^*_{\pm 0.16}/\mathbf{92.92_{\pm 0.19}}$ | $92.80^*_{\pm 0.16}/\mathbf{92.92_{\pm 0.19}}$ | $92.80^*_{\pm 0.16}/\mathbf{92.92_{\pm 0.19}}$ |
| | AdaS$^{0.9}$ | $93.71_{\pm 0.04}/\mathbf{94.09_{\pm 0.14}}$ | $93.75^*_{\pm 0.12}/\mathbf{94.05_{\pm 0.10}}$ | $93.75^*_{\pm 0.12}/\mathbf{94.05_{\pm 0.10}}$ | $93.75^*_{\pm 0.12}/\mathbf{94.05_{\pm 0.10}}$ | $93.75^*_{\pm 0.12}/\mathbf{94.05_{\pm 0.10}}$ |
| | AdaS$^{0.95}$ | $\mathbf{91.96_{\pm 0.35}}/91.89_{\pm 1.10}$ | $94.69_{\pm 0.15}/\mathbf{94.81_{\pm 0.16}}$ | $94.74^*_{\pm 0.16}/\mathbf{94.93_{\pm 0.11}}$ | $94.74^*_{\pm 0.16}/\mathbf{94.93_{\pm 0.11}}$ | $94.74^*_{\pm 0.16}/\mathbf{94.93_{\pm 0.11}}$ |
| | AdaS$^{0.975}$ | $87.99_{\pm 1.18}/\mathbf{88.05_{\pm 1.19}}$ | $93.35_{\pm 0.43}/\mathbf{93.46_{\pm 0.12}}$ | $94.74_{\pm 0.10}/\mathbf{94.77_{\pm 0.22}}$ | $94.88_{\pm 0.07}/\mathbf{95.07_{\pm 0.22}}$ | $94.94_{\pm 0.04}/\mathbf{95.14_{\pm 0.20}}$ |
| | RMSProp | $90.35_{\pm 1.21}/\mathbf{90.98_{\pm 0.36}}$ | $91.85_{\pm 0.19}/\mathbf{91.88_{\pm 0.47}}$ | $\mathbf{92.26_{\pm 0.16}}/92.25_{\pm 0.16}$ | $\mathbf{92.66_{\pm 0.12}}/92.39_{\pm 0.38}$ | $\mathbf{92.69_{\pm 0.33}}/92.62_{\pm 0.30}$ |
| | SLS | $\mathbf{93.30_{\pm 0.16}}/93.28_{\pm 0.10}$ | $93.41_{\pm 0.09}/\mathbf{93.48_{\pm 0.09}}$ | $93.39_{\pm 0.10}/\mathbf{93.49_{\pm 0.09}}$ | $93.34_{\pm 0.13}/\mathbf{93.41_{\pm 0.08}}$ | $93.33_{\pm 0.06}/\mathbf{93.45_{\pm 0.16}}$ |
| CIFAR100 | AdaBound | $\mathbf{70.15_{\pm 0.31}}/68.75_{\pm 0.42}$ | $\mathbf{72.29_{\pm 0.21}}/70.89_{\pm 0.26}$ | $\mathbf{72.91_{\pm 0.24}}/71.60_{\pm 0.40}$ | $\mathbf{73.10_{\pm 0.38}}/71.86_{\pm 0.18}$ | $\mathbf{73.16_{\pm 0.25}}/72.04_{\pm 0.30}$ |
| | AdaGrad | $66.07_{\pm 0.36}/\mathbf{66.12_{\pm 0.53}}$ | $\mathbf{67.39_{\pm 0.48}}/67.37_{\pm 0.39}$ | $\mathbf{67.62_{\pm 0.52}}/67.50_{\pm 0.57}$ | $\mathbf{67.85_{\pm 0.57}}/67.72_{\pm 0.31}$ | $67.75_{\pm 0.56}/\mathbf{67.76_{\pm 0.50}}$ |
| | AdaM | $68.32_{\pm 0.65}/\mathbf{68.51_{\pm 0.39}}$ | $69.33_{\pm 0.42}/\mathbf{69.56_{\pm 0.29}}$ | $69.60_{\pm 0.18}/\mathbf{70.07_{\pm 0.31}}$ | $69.72_{\pm 0.22}/\mathbf{70.49_{\pm 0.45}}$ | $70.09_{\pm 0.35}/\mathbf{70.34_{\pm 0.27}}$ |
| | AdaS$^{0.8}$ | $73.32_{\pm 0.30}/\mathbf{73.48_{\pm 0.12}}$ | $73.38^*_{\pm 0.28}/\mathbf{73.59_{\pm 0.09}}$ | $73.38^*_{\pm 0.28}/\mathbf{73.59_{\pm 0.09}}$ | $73.38^*_{\pm 0.28}/\mathbf{73.59_{\pm 0.09}}$ | $73.38^*_{\pm 0.28}/\mathbf{73.59_{\pm 0.09}}$ |
| | AdaS$^{0.9}$ | $\mathbf{75.24_{\pm 0.27}}/74.98_{\pm 0.15}$ | $\mathbf{75.27^*_{\pm 0.28}}/75.15_{\pm 0.17}$ | $\mathbf{75.27^*_{\pm 0.28}}/75.15_{\pm 0.17}$ | $\mathbf{75.27^*_{\pm 0.28}}/75.15_{\pm 0.17}$ | $\mathbf{75.27^*_{\pm 0.28}}/75.15_{\pm 0.17}$ |
| | AdaS$^{0.95}$ | $\mathbf{74.40_{\pm 0.17}}/73.86_{\pm 0.17}$ | $76.44_{\pm 0.30}/\mathbf{76.32_{\pm 0.33}}$ | $76.44_{\pm 0.34}/\mathbf{76.47_{\pm 0.31}}$ | $76.49^*_{\pm 0.37}/\mathbf{76.53_{\pm 0.30}}$ | $76.49^*_{\pm 0.37}/\mathbf{76.53_{\pm 0.30}}$ |
| | AdaS$^{0.975}$ | $63.62_{\pm 1.70}/\mathbf{64.44_{\pm 0.99}}$ | $\mathbf{74.21_{\pm 0.25}}/73.79_{\pm 0.54}$ | $76.59_{\pm 0.19}/\mathbf{77.05_{\pm 0.10}}$ | $76.62_{\pm 0.14}/\mathbf{77.15_{\pm 0.19}}$ | $76.68_{\pm 0.18}/\mathbf{77.23_{\pm 0.09}}$ |
| | RMSProp | $66.29_{\pm 0.42}/\mathbf{66.52_{\pm 1.41}}$ | $68.48_{\pm 0.31}/\mathbf{68.89_{\pm 0.43}}$ | $68.88_{\pm 0.20}/\mathbf{69.10_{\pm 0.36}}$ | $69.29_{\pm 0.55}/\mathbf{69.78_{\pm 0.46}}$ | $69.28_{\pm 0.50}/\mathbf{70.08_{\pm 0.23}}$ |
| | SLS | $\mathbf{66.81_{\pm 1.25}}/64.88_{\pm 1.23}$ | $\mathbf{73.48_{\pm 0.24}}/73.02_{\pm 0.20}$ | $\mathbf{73.78_{\pm 0.19}}/73.14_{\pm 0.21}$ | $\mathbf{73.74_{\pm 0.32}}/73.24_{\pm 0.11}$ | $\mathbf{73.77_{\pm 0.12}}/73.22_{\pm 0.11}$ |

Table 7: List of ResNeXt50 top-1 test accuracies using various optimizers and applied to various datasets. Note that each cell reports the average accuracy over each experimental trial, sub-scripted by the standard deviation over the trials. Values on the left are from trials trained using the initial learning rate as generated by autoHyper. Values on the right are from trials trained using the suggested initial learning rate. Note a '*' indicates that early-stop has been activated.

| Dataset | Optimizer | Epoch 50 | Epoch 100 | Epoch 150 | Epoch 200 | Epoch 250 |
|---|---|---|---|---|---|---|
| CIFAR10 | AdaBound | $88.71_{\pm1.28}/\mathbf{89.01_{\pm0.55}}$ | $\mathbf{90.90_{\pm0.25}}/90.75_{\pm0.39}$ | $90.76_{\pm0.65}/\mathbf{91.23_{\pm0.24}}$ | $\mathbf{91.63_{\pm0.30}}/91.55_{\pm0.29}$ | $\mathbf{91.69_{\pm0.33}}/91.42_{\pm0.42}$ |
| | AdaGrad | $87.80_{\pm1.32}/\mathbf{87.85_{\pm0.91}}$ | $89.25_{\pm0.33}/\mathbf{89.37_{\pm0.23}}$ | $\mathbf{89.79_{\pm0.27}}/89.63_{\pm0.69}$ | $\mathbf{90.07_{\pm0.27}}/89.80_{\pm0.11}$ | $\mathbf{90.13_{\pm0.19}}/90.07_{\pm0.27}$ |
| | AdaM | $\mathbf{90.11_{\pm0.59}}/89.74_{\pm0.89}$ | $91.15_{\pm0.19}/\mathbf{91.28_{\pm0.45}}$ | $91.68_{\pm0.26}/\mathbf{91.82_{\pm0.27}}$ | $91.61_{\pm0.24}/\mathbf{92.07_{\pm0.17}}$ | $92.12_{\pm0.07}/\mathbf{92.18_{\pm0.31}}$ |
| | AdaS$^{0.8}$ | $91.51_{\pm0.22}/\mathbf{91.56_{\pm0.13}}$ | $91.49^*_{\pm0.16}/\mathbf{91.56_{\pm0.07}}$ | $91.49^*_{\pm0.16}/\mathbf{91.56_{\pm0.07}}$ | $91.49^*_{\pm0.16}/\mathbf{91.56_{\pm0.07}}$ | $91.49^*_{\pm0.16}/\mathbf{91.56_{\pm0.07}}$ |
| | AdaS$^{0.9}$ | $93.31_{\pm0.15}/\mathbf{93.37_{\pm0.11}}$ | $93.51^*_{\pm0.12}/\mathbf{93.60_{\pm0.16}}$ | $93.51^*_{\pm0.12}/\mathbf{93.60_{\pm0.16}}$ | $93.51^*_{\pm0.12}/\mathbf{93.60_{\pm0.16}}$ | $93.51^*_{\pm0.12}/\mathbf{93.60_{\pm0.16}}$ |
| | AdaS$^{0.95}$ | $89.03_{\pm0.45}/\mathbf{90.49_{\pm0.24}}$ | $94.57_{\pm0.13}/\mathbf{94.61_{\pm0.15}}$ | $94.59_{\pm0.14}/\mathbf{94.62_{\pm0.10}}$ | $94.61^*_{\pm0.11}/\mathbf{94.62_{\pm0.10}}$ | $94.61^*_{\pm0.11}/\mathbf{94.62_{\pm0.10}}$ |
| | AdaS$^{0.975}$ | $\mathbf{87.65_{\pm0.55}}/85.33_{\pm1.38}$ | $92.07_{\pm0.48}/\mathbf{92.29_{\pm0.54}}$ | $\mathbf{94.93_{\pm0.07}}/94.86_{\pm0.11}$ | $\mathbf{95.03_{\pm0.09}}/94.98_{\pm0.13}$ | $95.02_{\pm0.06}/\mathbf{95.03_{\pm0.12}}$ |
| | RMSProp | $\mathbf{89.26_{\pm0.42}}/89.11_{\pm0.89}$ | $89.65_{\pm0.89}/\mathbf{90.17_{\pm0.30}}$ | $91.02_{\pm0.36}/\mathbf{91.63_{\pm0.13}}$ | $90.96_{\pm0.45}/\mathbf{91.80_{\pm0.17}}$ | $91.34_{\pm0.59}/\mathbf{92.15_{\pm0.20}}$ |
| | SLS | $93.05_{\pm0.31}/\mathbf{93.18_{\pm0.11}}$ | $\mathbf{93.52_{\pm0.12}}/93.40_{\pm0.10}$ | $\mathbf{93.51_{\pm0.20}}/93.50_{\pm0.12}$ | $\mathbf{93.54_{\pm0.22}}/93.47_{\pm0.12}$ | $\mathbf{93.56_{\pm0.20}}/93.49_{\pm0.14}$ |
| CIFAR100 | AdaBound | $66.66_{\pm0.64}/\mathbf{66.75_{\pm0.31}}$ | $69.92_{\pm0.40}/\mathbf{70.13_{\pm0.40}}$ | $70.63_{\pm0.23}/\mathbf{71.06_{\pm0.51}}$ | $71.02_{\pm0.28}/\mathbf{71.35_{\pm0.26}}$ | $71.20_{\pm0.34}/\mathbf{71.43_{\pm0.30}}$ |
| | AdaGrad | $\mathbf{63.75_{\pm0.57}}/62.97_{\pm0.54}$ | $\mathbf{65.27_{\pm0.46}}/65.01_{\pm0.64}$ | $\mathbf{65.36_{\pm0.48}}/65.30_{\pm0.41}$ | $\mathbf{65.75_{\pm0.41}}/65.65_{\pm0.37}$ | $\mathbf{66.03_{\pm0.56}}/65.66_{\pm0.36}$ |
| | AdaM | $\mathbf{66.74_{\pm0.46}}/66.47_{\pm0.45}$ | $68.40_{\pm0.28}/\mathbf{68.87_{\pm0.50}}$ | $68.37_{\pm0.53}/\mathbf{69.59_{\pm0.26}}$ | $69.04_{\pm0.18}/\mathbf{70.14_{\pm0.35}}$ | $69.12_{\pm0.16}/\mathbf{70.32_{\pm0.46}}$ |
| | AdaS$^{0.8}$ | $72.06_{\pm0.41}/\mathbf{72.39_{\pm0.21}}$ | $72.00^*_{\pm0.44}/\mathbf{72.41_{\pm0.16}}$ | $72.00^*_{\pm0.44}/\mathbf{72.41_{\pm0.16}}$ | $72.00^*_{\pm0.44}/\mathbf{72.41_{\pm0.16}}$ | $72.00^*_{\pm0.44}/\mathbf{72.41_{\pm0.16}}$ |
| | AdaS$^{0.9}$ | $74.23_{\pm0.28}/\mathbf{74.30_{\pm0.11}}$ | $74.41^*_{\pm0.26}/\mathbf{74.43_{\pm0.14}}$ | $74.41^*_{\pm0.26}/\mathbf{74.43_{\pm0.14}}$ | $74.41^*_{\pm0.26}/\mathbf{74.43_{\pm0.14}}$ | $74.41^*_{\pm0.26}/\mathbf{74.43_{\pm0.14}}$ |
| | AdaS$^{0.95}$ | $\mathbf{72.78_{\pm0.09}}/72.29_{\pm0.37}$ | $75.47_{\pm0.10}/\mathbf{76.03_{\pm0.34}}$ | $75.74_{\pm0.03}/\mathbf{75.95_{\pm0.26}}$ | $75.63^*_{\pm0.12}/\mathbf{75.95_{\pm0.26}}$ | $75.63^*_{\pm0.12}/\mathbf{75.95_{\pm0.26}}$ |
| | AdaS$^{0.975}$ | $63.16_{\pm1.84}/\mathbf{63.67_{\pm1.48}}$ | $74.42_{\pm0.47}/\mathbf{74.52_{\pm0.44}}$ | $76.12_{\pm0.17}/\mathbf{76.27_{\pm0.23}}$ | $76.45_{\pm0.16}/\mathbf{76.47_{\pm0.18}}$ | $\mathbf{76.58_{\pm0.21}}/76.46_{\pm0.24}$ |
| | RMSProp | $63.69_{\pm0.66}/\mathbf{63.96_{\pm1.40}}$ | $66.24_{\pm0.28}/\mathbf{67.82_{\pm0.82}}$ | $66.35_{\pm0.26}/\mathbf{67.97_{\pm1.16}}$ | $66.83_{\pm0.39}/\mathbf{69.31_{\pm0.27}}$ | $67.17_{\pm0.70}/\mathbf{69.45_{\pm1.17}}$ |
| | SLS | $55.34_{\pm4.77}/\mathbf{58.80_{\pm3.56}}$ | $70.10_{\pm1.18}/\mathbf{70.75_{\pm0.98}}$ | $71.51_{\pm0.31}/\mathbf{71.73_{\pm0.38}}$ | $71.89_{\pm0.09}/\mathbf{72.03_{\pm0.40}}$ | $71.82_{\pm0.22}/\mathbf{72.08_{\pm0.43}}$ |

Table 8: List of DenseNet121 top-1 test-accuracies using various optimizers and applied to various datasets. Note that each cell reports the average accuracy over each experimental trial, sub-scripted by the standard deviation over the trials. Values on the left are from trials trained using the initial learning rate as generated by autoHyper. Values on the right are from trials trained using the suggested initial learning rate. Note a '*' indicates that early-stop has been activated.

| Dataset | Optimizer | Epoch 50 | Epoch 100 | Epoch 150 | Epoch 200 | Epoch 250 |
|---|---|---|---|---|---|---|
| CIFAR10 | AdaBound | $88.57_{\pm0.49}/\mathbf{89.73_{\pm0.32}}$ | $90.09_{\pm0.23}/\mathbf{90.64_{\pm0.23}}$ | $90.65_{\pm0.24}/\mathbf{91.20_{\pm0.09}}$ | $90.59_{\pm0.46}/\mathbf{91.32_{\pm0.12}}$ | $90.96_{\pm0.48}/\mathbf{91.65_{\pm0.20}}$ |
| | AdaGrad | $\mathbf{88.57_{\pm0.29}}/88.31_{\pm0.45}$ | $\mathbf{89.20_{\pm0.23}}/89.03_{\pm0.27}$ | $\mathbf{89.36_{\pm0.09}}/89.26_{\pm0.25}$ | $\mathbf{89.47_{\pm0.12}}/89.33_{\pm0.25}$ | $\mathbf{89.85_{\pm0.19}}/89.52_{\pm0.16}$ |
| | AdaM | $\mathbf{90.75_{\pm0.28}}/89.42_{\pm0.72}$ | $\mathbf{91.39_{\pm0.32}}/90.28_{\pm0.49}$ | $\mathbf{91.75_{\pm0.27}}/90.84_{\pm0.16}$ | $\mathbf{91.72_{\pm0.28}}/91.29_{\pm0.26}$ | $\mathbf{91.86_{\pm0.26}}/91.32_{\pm0.43}$ |
| | AdaS$^{0.8}$ | $\mathbf{91.61_{\pm0.20}}/91.28_{\pm0.26}$ | $91.59^*_{\pm0.25}/91.28_{\pm0.23}$ | $91.59^*_{\pm0.25}/91.28_{\pm0.23}$ | $91.59^*_{\pm0.25}/91.28_{\pm0.23}$ | $91.59^*_{\pm0.25}/91.28_{\pm0.23}$ |
| | AdaS$^{0.9}$ | $\mathbf{92.91_{\pm0.08}}/92.89_{\pm0.20}$ | $92.97^*_{\pm0.18}/\mathbf{93.06_{\pm0.14}}$ | $92.97^*_{\pm0.18}/\mathbf{93.06_{\pm0.14}}$ | $92.97^*_{\pm0.18}/\mathbf{93.06_{\pm0.14}}$ | $92.97^*_{\pm0.18}/\mathbf{93.06_{\pm0.14}}$ |
| | AdaS$^{0.95}$ | $91.59_{\pm0.36}/\mathbf{91.59_{\pm0.23}}$ | $93.33_{\pm0.35}/\mathbf{93.45_{\pm0.18}}$ | $93.33^*_{\pm0.24}/\mathbf{93.51_{\pm0.20}}$ | $93.33^*_{\pm0.24}/\mathbf{93.51_{\pm0.20}}$ | $93.33^*_{\pm0.24}/\mathbf{93.51_{\pm0.20}}$ |
| | AdaS$^{0.975}$ | $\mathbf{86.96_{\pm1.15}}/86.17_{\pm0.80}$ | $91.87_{\pm0.26}/\mathbf{92.06_{\pm0.51}}$ | $93.17_{\pm0.20}/\mathbf{93.53_{\pm0.19}}$ | $93.40_{\pm0.22}/\mathbf{93.72_{\pm0.15}}$ | $93.47_{\pm0.24}/\mathbf{93.83_{\pm0.20}}$ |
| | RMSProp | $\mathbf{88.80_{\pm0.93}}/88.19_{\pm1.18}$ | $\mathbf{90.95_{\pm0.85}}/89.76_{\pm1.15}$ | $\mathbf{91.54_{\pm0.52}}/90.77_{\pm0.64}$ | $\mathbf{91.65_{\pm0.36}}/90.95_{\pm0.48}$ | $\mathbf{91.83_{\pm0.30}}/91.29_{\pm0.20}$ |
| | SLS | $\mathbf{92.95_{\pm0.28}}/92.78_{\pm0.07}$ | $\mathbf{93.35_{\pm0.20}}/93.14_{\pm0.16}$ | $\mathbf{93.40_{\pm0.17}}/93.12_{\pm0.15}$ | $\mathbf{93.38_{\pm0.15}}/93.16_{\pm0.11}$ | $\mathbf{93.36_{\pm0.18}}/93.16_{\pm0.13}$ |
| CIFAR100 | AdaBound | $63.87_{\pm0.25}/\mathbf{65.30_{\pm0.28}}$ | $65.61_{\pm0.18}/\mathbf{68.00_{\pm0.30}}$ | $66.66_{\pm0.22}/\mathbf{68.65_{\pm0.21}}$ | $66.69_{\pm0.37}/\mathbf{68.46_{\pm0.32}}$ | $67.00_{\pm0.20}/\mathbf{68.90_{\pm0.36}}$ |
| | AdaGrad | $\mathbf{61.62_{\pm0.46}}/61.08_{\pm0.28}$ | $\mathbf{62.01_{\pm0.29}}/61.43_{\pm0.31}$ | $\mathbf{62.40_{\pm0.50}}/61.86_{\pm0.50}$ | $\mathbf{62.91_{\pm0.46}}/62.09_{\pm0.22}$ | $\mathbf{62.79_{\pm0.35}}/62.14_{\pm0.15}$ |
| | AdaM | $\mathbf{66.53_{\pm0.30}}/64.50_{\pm0.33}$ | $\mathbf{67.64_{\pm0.35}}/65.69_{\pm0.29}$ | $\mathbf{68.32_{\pm0.27}}/66.42_{\pm0.57}$ | $\mathbf{68.46_{\pm0.28}}/66.89_{\pm0.53}$ | $\mathbf{69.05_{\pm0.49}}/67.48_{\pm0.17}$ |
| | AdaS$^{0.8}$ | $\mathbf{70.90_{\pm0.11}}/70.59_{\pm0.39}$ | $\mathbf{71.01^*_{\pm0.28}}/70.63_{\pm0.33}$ | $\mathbf{71.01^*_{\pm0.28}}/70.63_{\pm0.33}$ | $\mathbf{71.01^*_{\pm0.28}}/70.63_{\pm0.33}$ | $\mathbf{71.01^*_{\pm0.28}}/70.63_{\pm0.33}$ |
| | AdaS$^{0.9}$ | $73.11_{\pm0.49}/\mathbf{73.13_{\pm0.30}}$ | $73.13_{\pm0.44}/\mathbf{73.25_{\pm0.25}}$ | $73.13_{\pm0.44}/\mathbf{73.25_{\pm0.25}}$ | $73.13_{\pm0.44}/\mathbf{73.25_{\pm0.25}}$ | $73.13_{\pm0.44}/\mathbf{73.25_{\pm0.25}}$ |
| | AdaS$^{0.95}$ | $72.62_{\pm0.50}/\mathbf{72.90_{\pm0.72}}$ | $73.98_{\pm0.21}/\mathbf{74.09_{\pm0.30}}$ | $73.98^*_{\pm0.33}/\mathbf{74.22_{\pm0.24}}$ | $73.98^*_{\pm0.33}/\mathbf{74.22_{\pm0.24}}$ | $73.98^*_{\pm0.33}/\mathbf{74.22_{\pm0.24}}$ |
| | AdaS$^{0.975}$ | $61.85_{\pm1.59}/\mathbf{63.44_{\pm1.90}}$ | $72.16_{\pm0.56}/\mathbf{72.65_{\pm0.85}}$ | $73.81_{\pm0.44}/\mathbf{73.92_{\pm0.52}}$ | $73.88_{\pm0.42}/\mathbf{73.99_{\pm0.45}}$ | $73.97_{\pm0.36}/\mathbf{74.10_{\pm0.47}}$ |
| | RMSProp | $\mathbf{64.73_{\pm0.00}}/62.80_{\pm0.54}$ | $\mathbf{66.22_{\pm0.00}}/64.85_{\pm0.28}$ | $\mathbf{67.57_{\pm0.00}}/65.74_{\pm0.40}$ | $\mathbf{67.35_{\pm0.00}}/66.30_{\pm0.42}$ | $\mathbf{68.13_{\pm0.00}}/66.61_{\pm0.58}$ |
| | SLS | $52.01_{\pm7.25}/\mathbf{55.66_{\pm6.26}}$ | $\mathbf{65.96_{\pm1.20}}/62.29_{\pm5.97}$ | $\mathbf{69.88_{\pm0.24}}/67.28_{\pm1.15}$ | $\mathbf{70.12_{\pm0.45}}/68.63_{\pm1.61}$ | $\mathbf{70.25_{\pm0.19}}/69.44_{\pm0.61}$ |

# D    COMPARISON AGAINST STATE OF THE ART (RANDOM SEARCH)

## D.1    SETUP

The search space is set to $[1 \times 10^{-4}, 0.1]$ and a *loguniform* (see SciPy) distribution is used for sampling. This is motivated by the fact that autoHyper also uses and logarithmically-spaced grid space. We note that we ran initial tests against a uniform distribution for sampling was done and showed slightly worse results, as the favouring of smaller learning rates benefits the optimizers we considered. In keeping with autoHyper's design, the learning rate that resulted in lowest training loss after 5 epochs was chosen. One could also track validation accuracy, however as visualized in Figures 5(a) & 13, validation loss is more stable for the datasets we are considering. This selection could be altered if the dataset being used exhibits a different behaviour, however this would be a manual alteration at the selection of the practitioner – one that does not need to be made if using autoHyper.

## D.2    ADDITIONAL DISCUSSION AND RESULTS

**Can you replace $\mathcal{Z}(\eta)$ with validation loss?** Replacing $\mathcal{Z}(\eta)$ with validation loss does not work because greedily taking validation loss (or accuracy) is not stable nor domain independent. Analyzing Figures 5(a) & 9, validation loss/accuracy is unstable since either the network (EfficientNetB0 in Figure 9) or the dataset (TinyImageNet in Figure 5(a)) results in unstable top-1 test accuracy/test loss scores that are unreliable to track. See also Figure 14, which demonstrates the inbaility to track validation loss/accuracy for various learning rates. Further, validation accuracy/loss can vary greatly based on initialization, whereas our method does not vary due to its low-rank factorization. Finally, our metric, $\mathcal{Z}(\eta)$, is always guaranteed to be zero with a sufficiently small learning rate and maximized with large learning rates, therefore we can always dynamically adapt our search range to the proper range. This fact is not so true for tracking validation accuracy/loss.

Additionally, low validation loss does not correlate to high validation accuracy (an additional figure, Figure 12, in Appendix C shows this). You might then suggest to take a $k$ of the best performing learning rates based on validation accuracy/loss and focus on those, but this requires you to manually define $k$ then attempt a manually defined Grid/Random Search refinements around those areas, with manual heuristics to indicate when to stop searching, whereas our method is fully automatic and self-converges. Not to mention, this would take more time.

In summation, existing SOTA method like Random Search cannot compete with autoHyper when given similar budgets and minimizing the manual intervention/refinement. This displays autoHyper's prominent feature of being a low-cost, fully automatic algorithm to search for optimal hyper-parameter bounds (namely in this work, the initial learning rate). Future work could include using autoHyper to quickly discover this optimal hyper-parameter range, and then further refine using more extensive HPO methods with greater budgets if truly superior performance is required, and this could further alleviate a lot of manual refinement that currently plagues existing SOTA methods.

Table 9: Learning rates for ResNet34 Random Search comparison. Left inner columns show Random Search generated, right inner columns show autoHyper generated.

| Optimizer | CIFAR10 | | CIFAR100 | | TinyImageNet | |
|---|---|---|---|---|---|---|
| AdaM | 0.000330 | 0.000333 | 0.000125 | 0.000241 | 0.000175 | 0.0001965 |
| AdaBound | 0.000392 | 0.000347 | 0.000353 | 0.000347 | 0.000124 | 0.0000944 |
| AdaGrad | 0.001598 | 0.002861 | 0.001828 | 0.002236 | 0.000715 | 0.0022359 |
| AdaS$^{(0.9)}$ | 0.006779 | 0.012374 | 0.009252 | 0.010190 | 0.039480 | 0.0085857 |

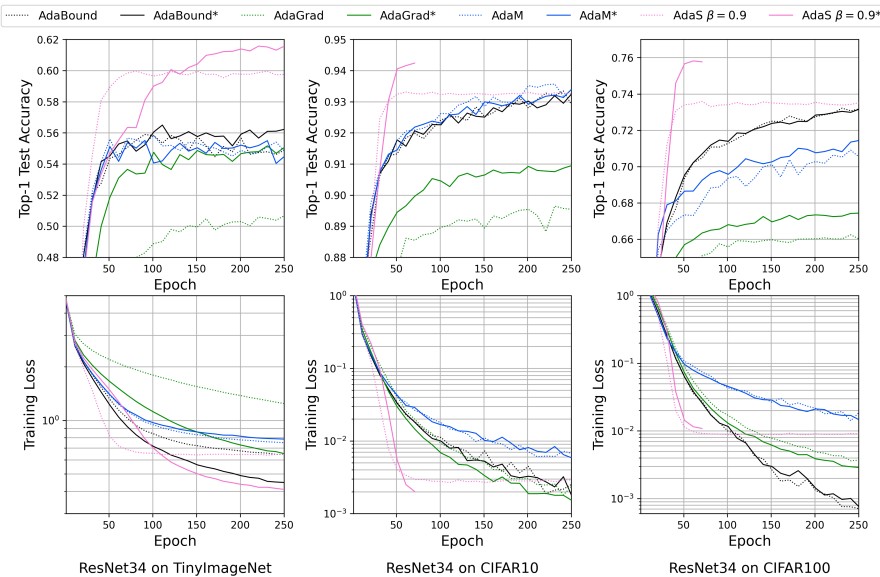

Figure 13: Top-1 test accuracy and train loss for ResNet34 applied to TinyImageNet, CIFAR10, and CIFAR100, using learning rates as suggested by either a Random Search (as described above) or autoHyper. Titles below plots indicate what experiment the above plots refers to. Legend labels marked by '*' (solid lines) show results for autoHyper generated learning rates and dotted lines are the Random Search results.

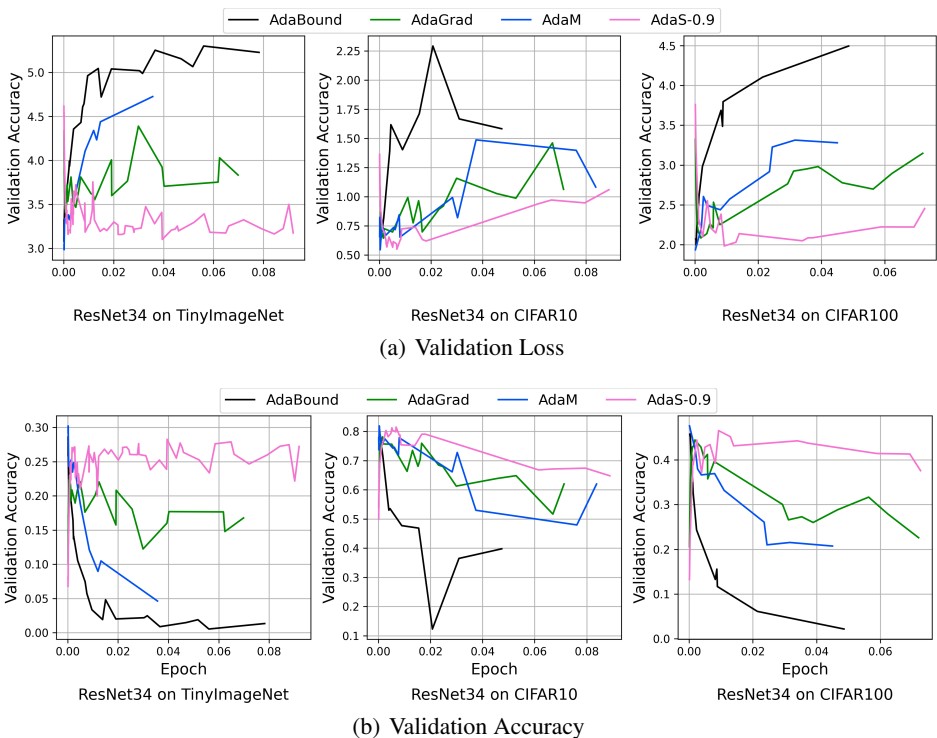

Figure 14: Visualization of the (a) validation loss and (b) validation accuracy for various learning rates ResNet34 on various datasets. These figures demonstrate the inability to propoerly track these metrics as we do ours (*i.e.* $\mathcal{Z}(\eta)$)

