# OpenReview forum: "Response Modeling of Hyper-Parameters for Deep Convolutional Neural Networks"
_ICLR.cc/2021/Conference — Reject_

### Official Review · AnonReviewer3 · 2020-10-26
**A new algorithm to choose the hyperparameter - initial learning rate, but has some feasibility and correctness problem**

**Rating:** 5
**Confidence:** 3

**Review:**

###############################################################

Summary:

This paper used the knowledge gain to provide an algorithm to choose the initial learning rate. The paper explained the reason that choosing the specific response function and demonstrated the effectiveness of the algorithm by several experiments.

##############################################################

Reason for Score:

The intuition is reasonable, but there are several points regarding the feasibility and correctness that need to be clarified.

##############################################################

pros:

1, This paper proposed a dynamic tracking algorithm of low computational overhead on the order of minutes and hours to find a good initial learning rate.

2, The paper explained in detail about the reason to choose the specific response surface function, and demonstrated the reasonability of the algorithm.

##############################################################

cons:

1, The algorithm had an implicit assumption: the response function is monotonously decreasing (maybe under expectation). Otherwise, the algorithm can either not converge or go to a local minimum. Even though the author list some examples, I doubt about this. Imagine that we choose a very large learning rate (some unreasonable number that just for a counterexample), the training loss cannot be decreased, then I doubt that the response function cannot decrease either.  Did the author conduct some examples about this? If this is the case, then how to choose the range of the searching area matters a lot.

2, The algorithm induces some "hyperparameter" again. Such as the alpha, and the condition "rate of change plateaus". How to choose the alpha, and what is the "rate of change plateaus"? Does these "hyperparameters" influence the result?

3, The initial learning rate seems only influence the convergence rate for training loss, but why it influences the testing accuracy? I suggest the author at least give some reasonable explanation about this.

4, The paper compared the algorithm only with the baseline. Is there any other work about tuning the initial learning rate? Or this paper is the first?

5, minor problems:

(1) In equation 1, x is not showed in expectation function (I can only see X(train)), and also, it should be the minimum of expectation, not the expectation of minimum.

(2) In algorithm 1, there is nowhere that the index i is changed.

(3) I suggest the author change the color of the lines in graph. It is very difficult to find which line corresponds to which experiment.

Thanks for your rebuttal. It solves some of my concerns. However, combining with rebuttal and other reviewers' comment, I think only choosing the initial learning rate is not that reasonable. It may be more convincing to me that the whole parameters are chosen together.

---

> ### Author Response · Authors · 2020-11-14
> **Author Response**
>
> Thank you for taking the time and effort to review our paper and we appreciate your extensive comments and insight. To address the reviewer’s points...
>
> ** Note draft amendments are highlighted in red.
>
> ### “The algorithm had an implicit ...the range of the searching area matters a lot”
>
> The reviewer is correct that our response function is not monotonously decreasing, however we note that we do not assume this: lack of explanation on our part has caused this confusion. We highlight in Figure 4 that our response surface is not monotonically decreasing, however since $\mathcal{Z}$ lies between 0 and 1, its cumulative product of sequential learning rates is. This is visualized in Figure 4, the orange line. Therefore, we designed our algorithm to track this cumulative product to ensure monotonicity and ensure convergence. Importantly, we note that it is important to start at a small learning rate. $\mathcal{Z}$ will equal 1 (meaning the low-rank factorized weight matrix has complete zero-valued singular values) for a very small learning rates as each layer’s weight matrix will be fully populated with noise (see response to reviewer #2). Therefore, if we start sufficiently small, there is only progress to be made. Additionally, the reviewer is correct that the choice of searching range matters a lot, however our method is not constrained to its initial search range. If initially, $\mathcal{Z}$ is too low, then our algorithm will reduce its minimum search bound and restart (see Algorithm 1, line 18-19). Similarly, if our algorithm reaches the end of its search bounds and has not yet converged, it will increase this range and continue to track the cumulative product of $\mathcal{Z}$ until it does converge, therefore our method is not subject to initial choice of searching range, so long as it is granular enough. Note that a coarser grid space can still work, only the converge time will be longer.
>
> ### The algorithm introduces some “hyper-parameter” again  …influence result?”
>
> The reviewer is correct in that alpha is technically a hyper-parameter, however it acts as stopping condition in our algorithm to prevent infinite recursion. Our algorithm will continue to “contract” its search space until the minimum and maximum value of the search space are not significantly different. Alpha (set to 5e-5), is threshold for significant difference; if our search space bounds differ by a value <5e-5, then our search stops. Naturally, changing this threshold to something large would heavily affect results, however it would not make sense, as then your search bounds are in fact going to be significantly different. One can choose a smaller value, this will simply increase convergence time.  As for “rate of change plateaus”, this is not a hyper-parameter, but rather the solution to our proposed response surface. We refer the reader to subsection 2.1 for a more detailed explanation, but note that our response surface is modelled such that optimal learning rates lie when the rate of change (cumulative product) or our $\mathcal{Z}$ plateaus
>
> ### The initial learning rate seems to … reasonable explanation about this”
>
> We believe the reviewer is referring here to Figure 1(a), and we note that this figure simply shows the auxiliary behaviour or our response model and that optimizing against our response model also minimizes training loss, and in our experimental results, we show how this further extends to testing accuracy in its competitiveness. Additionally, we note that in general, anything that shows training loss convergence and minimization corresponds to a high testing accuracy performance (Wilson 2017 discusses similar notes, although this is a generally accepted notion). Additionally, as noted in our introduction, the optimization of the initial learning rate has a critical influence on testing accuracy (Goodfellow et al., 2016; Bergstra & Bengio, 2012; Yu & Zhu, 2020).
>
> ### “The paper compare the...or this paper is the first?”
>
> Hyper-parameter tuning is not new however, as far as we know, this is the first method that (1) introduces a well-defined response surface and (2) tunes the initial learning rate automatically with no manual intervention, no human intuition to set initial searching space, and at orders of magnitude faster than previous methods such as grid searching or bayesian optimization. We refer the reviewer to Figure 3 that shows the number of trialed learning rates, and the total time consumed for each trial on a single RTX2080 ti. Note that each trial only consumes 5 epochs of training as well. The number of epochs consumed in the optimization phase of our method is, in all cases but a few, less than the total number of epochs required for one full training cycle (See Figure 3).
>
> ### Minor problems
>
> We thank the reviewer for identifying these issues, which have now been addressed in the latest iteration to our draft.
>
> ** Note also that our anonymized code has been uploaded if the reviewers wish to examine it.

---

### Official Review · AnonReviewer4 · 2020-10-27
**Interesting algorithm but lacking competitive baselines**

**Rating:** 4
**Confidence:** 4

**Review:**

Overview:
Overall I find this an interesting algorithm, but have several serious concerns about the (lack of) experimental baselines.

My primary concern is that, given this paper is introducing a new hyperparameter tuning algorithm, there are no comparisons to baseline hyperparameter tuning setups. For example, in Figure 3, what is the best final validation and/or test accuracies achieved by autoHyper and by random search, for the same number of trials (one could even run autoHyper first to see how many trials it takes, and then see if random search could beat it in that many trials). I understand that BayesOpt algorithms can require more involved engineering to set up, but there do exist open source codebases that could be used (such as https://github.com/HIPS/Spearmint), and for a very solid paper I believe that comparing to existing SOTA methods would also be useful. For all of these methods, one could take the validation loss at 5 epochs (as is done in the proposed method) as the metric to tune on for a fair comparison.

Additionally, the suggested initial LRs seem problematic to compare to. It would useful to highlight how the suggested initial LRs were tuned. Using the information in Appendix D from Wilson 2017, I see they were tuned using less trials than the proposed algorithm, and using a grid search algorithm that may be worse than random search (Bergstra & Bengio (2012)). Furthermore, these suggested initial LRs were for a different model (this VGG model referenced from Wilson 2017 http://torch.ch/blog/2015/07/30/cifar.html) than the models considered in the experiments here, which means they could serve as a poor baseline. This baseline weakness is also shown when one considers that the ranges of values tuned over in Wilson 2017 seem to be quite close to several of the values proposed by autoHyper, meaning that perhaps repeating the same grid search (even with the same early stopping as in autoHyper) could be competitive (for Adam, Wilson 2017 considered {0.005, 0.001, 0.0005, 0.0003, 0.0001, 0.00005} and autoHyper proposed 0.000333, and for AdaGrad Wilson 2017 considered {0.1, 0.05, 0.01, 0.0075, 0.005} and autoHyper proposed 0.0049724). In the cases where the autoHyper values are not close, they are sometimes (although not always) outside the range considered by Wilson 2017, which could bias the results towards the proposed algorithm because the suggested LRs may not have had the chance to encounter the more successful values proposed by autoHyper (in the case of AdaGrad, Wilson 2017 considered {0.1, 0.05, 0.01, 0.0075, 0.005} while the proposed method tunes within [1e-4., 0.1] and selects 0.002861 for CIFAR10 ResNet34). Several recent works have shown that when tuning optimizers one needs to be careful to report the ranges used, as changing the hyperparameter ranges can drastically affect experimental results (https://arxiv.org/abs/2007.01547, https://arxiv.org/abs/1910.11758, https://arxiv.org/abs/1910.05446), and while they mainly consider the effect on optimizer comparisons the point still stands for comparisons between hyperparameter tuning algorithms (when comparing to random search, the range of LRs should be the same for each tuning algorithm).

Pros:
-It is useful to show that the tuning algorithm works across many optimizers, models, and datasets. The types of experiments seem sufficient, just missing baseline tuning algorithms.
-It is important to call out the negative results presented in experiments, and the authors did a good job of that (when applicable) in section 4.2

Concerns:
-Figure 1a would be much more informative if you showed the entire training trajectory, including past the first 5 epochs, to see if the selected learning rates actually generalize noticeably better.
-In your conclusion you discuss that you could extend your tuning algorithm to multiple hyperparameters, and I believe in order to truly demonstrate its capabilities this would be required, given that relatively simple baselines such as random search can continue to perform well in multi-dimensional tuning setups (although I would recommend against tuning the batch size given that it interacts so strongly with numerous other hyperparameters, and would instead suggest tuning momentum, weight decay, and/or label smoothing).

Writing:
-Is there an extra 2 in the denominator of Zt(λ)?
-It was initially confusing that the gradient notation was used for the constraint on Eq. 5, it would be clearer if the authors stated earlier that they did not compute the literal gradients (which would require backpropagating through unrolled updates).
-The “Observations on the generalization characteristics of optimizers.” subsection seems somewhat out of place

Additional feedback, comments, suggestions for improvement and questions for the authors:
-Instead of calculating Z(λ) with G¯λ(t, l), one could imagine making another baseline out of using validation accuracy at each epoch (perhaps normalized by some SOTA number if being in [0, 1] is desirable.)
-While very informative, Figures 5 and 10 seem cluttered, and in addition to them it may help for a future version to have the diff between the baseline and proposed method instead of two different lines

---

> ### Author Response · Authors · 2020-11-14
> **Author Response**
>
> Thank you for taking the time and effort to review our paper and we appreciate your extensive comments and insight. To address the reviewer’s points...
>
> ** Note draft amendments are highlighted in red.
>
> ### “My primary concern is that... tune on for a fair comparison”
>
> The reviewer raises a good point and we are making attempts at completing such experimentation; the draft will be updated soon. Initial findings (ResNet18 on CIFAR10 with AdaM + AdaGrad) using grid searching suggest that taking validation loss at 5 epochs (as suggested) is plausible, achieving competitive results. The important drawback here is the choice of initial grid space, which heavily affects results. In these initial experiments, we used a grid space that we know a priori  will include the “optimal” learning rate, when in practice, over different datasets/models/optimizers, this knowledge would not be present, thereby biasing the grid search’s performance. Our method dynamically grows or shrinks its own search space and is therefore not subject to manual intervention or need for a priori knowledge on an optimal grid space, and can fully autonomously tune the initial learning rate, which is its primary feature --No effort on the part of the researcher is required.
>
> ### “Additionally, the suggested initial ... they could sever as a poor baseline”
>
> Thank you for the references mentioned here, we have included some of them in the updated draft. We note that attempts at finding other work proposing alternative hyper-parameter settings for the experimental setups we ran was made; in absence of other work, original experimental settings were used. We understand the potential pitfalls of this method but extensive HPO for every experimental setup we ran was not possible, highlighting another benefit in the efficiency of our method.
>
> The original authors would have performed some form of HPO. On Wilson 2017’s use of different models, we found in our initial testing the learning rates suggested in that work were more optimal and we opted to use them as they were more competitive.
>
> ### “The baseline weakness …  autoHyper proposed 0.0049724“
>
> As the reviewer suggests, repeating the grid search around this 0.0003 value could achieve competitive results to our method, however this is exactly the benefit of our method; it will automatically find this optimal range or value without any effort from the practitioner, whereas repeating a grid search requires increasing manual effort, analysis, and intuition (and time). We identify this therefore as a strength of our method. If truly superior performance is desired, future work might include using our method to find this optimal 0.0003 value for example and then performing more intensive HPO around this range, all automatically, however this is not the intention of our work.
>
> ### “In the cases where the autoHyper ... same for each tuning algorithm”
>
> We argue that this does not bias the results towards our method but highlights its strength and attraction. Other methods not having the chance to encounter these more successful values is due to limitations in human effort to further refine their search, whereas our method automatically finds this range and considers it. We apologize and identify that lack of clarity on our part has led to this confusion and primary objective of our method, which is fundamentally different to SOTA HPO methods. Additionally, we agree that search range is critical to final performance as the reviewer suggests, but note that our method is free to grow or shrink the search bounds in relation to our response model.
>
> ### “Figure 1a would be more...entire training trajectory”
>
> We agree with the reviewer here however such visualization are not currently possible. Attempts at completing such experiments will be made prior to the November 24 deadline.
>
> ### “In your conclusion you discuss …  decay, and/or label smoothing”
>
> We agree with the reviewer and note that our future work includes such development. We also note our objective to lay the foundational proof-of-concept of this method in our currently proposed work, as is discussed in our response to reviewer #1 under [### “presented work would be more interesting...weight decay and convolution filter size and channels”]. To add to this, we note as in our conclusion that no matter how the hyper-parameter set dimension grows, we can fit a polynomial surface to that response model and, given some starting point, can follow along the tangent direction of the gradient to this polynomial and descend in a “line search” in that direction.
>
> ### Writing and additional feedback response
>
> We thank the reviewer for pointing out various issues here and have ammended the draft accordingly. A diff figure will follow in a subsequent draft update.
>
> ** Note also that our anonymized code has been uploaded if the reviewers wish to examine it.

---

> > ### Comment · AnonReviewer4 · 2020-11-20
> > **Validation accuracy baselines**
> >
> > Got it, that makes sense. Rereading the draft, I understand the point is to find better initial LR ranges, not tune the LR. However, I still don't see why one could not take your Algorithm 1 and rerun it, replacing Z(η_i) with the validation accuracy? Wouldn't this be cheaper to compute compared to your response function, and perhaps also be able to find a better initial LR range? Similarly to your proposed method, often times validation accuracy curves can change dramatically in the first 5 or so epochs and then plateau (similar to your Figures 7, 8)?
> >
> > Overall I believe the updated draft is an improvement over the original, and I thank the authors for their timely response and updates. Unfortunately I still don't believe a real apples-to-apples comparison has been made to more commonplace existing metrics (validation accuracy), so I don't see myself updating my score.

---

> > > ### Author Response · Authors · 2020-11-21
> > > **Author Response**
> > >
> > > Replacing Z(n_i) with validation accuracy does not work because greedily taking validation accuracy (or loss as initially suggested) is not stable nor domain independent. Before explaining this, we note that in our current experiments, using a randomized grid search taking the lowest validation loss after 5 epochs as you suggested, for AdaGrad on TinyImageNet, with the same number of trials as autoHyper, is over 4% worse in top-1 test accuracy for the selected learning rate. Note that if we instead tracked the validation accuracy here, the same learning rate would have been selected (which is 0.000715).
> > >
> > > Analyzing Figures 5 and 9, validation loss/accuracy is unstable since either the network (EfficientNetB0 in Figure 9) or the dataset (TinyImageNet in Figure 5) results in unstable top-1 test accuracy scores that are unreliable to track. Further, validation accuracy/loss can vary greatly based on initialization, whereas our method does not vary due to its low-rank factorization. Finally, as you mentioned, search range heavily influences results. Our metric, Z(n_i), is always guaranteed to be zero with a sufficiently small learning rate and maximized with large learning rates, therefore we can always dynamically adapt our search range to the proper range. This fact is not so true for tracking validation accuracy/loss.
> > >
> > > Additionally, low validation loss does not correlate to high validation accuracy (an additional figure, Figure 12, in Appendix C is added to show this). You might then suggest to take a “k” of the “best” performing learning rates based on validation accuracy/loss and focus on those, but this requires you to manually define k then attempt a manually defined grid search refinements around those areas, with manual heuristics to indicate when to stop searching, whereas our method is fully automatic and self-converges. Not to mention, this would take more time.

---

> ### Author Response · Authors · 2020-11-24
> **Random Search Results**
>
> We would like to notify the reviewer individually that our random search comparison results have now been uploaded to Appendix D, with them being moved to the main draft soon. Thank you.

---

### Official Review · AnonReviewer1 · 2020-10-29
**Review for Response Modeling of Hyper-Parameters for Deep Convolution Neural Network**

**Rating:** 4
**Confidence:** 3

**Review:**

In this paper, the authors propose a new hyper-parameter optimization method based on a new response function defined on the low-rank factorization of 4D convolution weights. The presented approach appears to mathematically solid and interesting. Although I am not an expert in this hyper-parameter optimization, I think the approach has the potential to speed up and improve the hyper-parameter search or neural network architecture search.

In my opinion, the demonstrated experiments are less interesting because the focus is only on selecting a single initial learning rate. This is not that interesting because we normally have a learning rate scheduler that will change over time. The experiment somehow ignores different types of learning rate schedulers for comparisons. In addition, as the learning rate is just a scalar, we can simply do some grid search such as [0.01, 0.001, ..] to coarsely find a good learning rate and then refine.

Also, the presented work would be more interesting if it can demonstrate improvement in other hyper-paramter optimization such as weight decay and convolution filter size and channels.

There are some questions about the presented approach:
- Are all the convolution layers in a CNN used in Eqn (4)? The definition of mode-3 and mode-4 is not clear to me.
-

---

> ### Author Response · Authors · 2020-11-14
> **Author Response**
>
> Thank you for taking the time and effort to review our paper and we appreciate your extensive comments and insight. To address the reviewer’s points...
>
> ** Note draft amendments are highlighted in red.
>
> ### Ignorance of scheduled learning rate: “the experiment somehow ignores...learning rate schedulers for comparison...”
>
> We initially shared your concerns, however our reasoning for ignoring scheduled learning rates is as follows: First, scheduled learning rates themselves introduce additional hyper-parameters (such as step-size and decay-rate) that we also need to be optimized, and are very sensitive to the initial learning rate. Such optimization would require manual tuning which goes against our method. Second, although it is common to have a scheduled learning rate, initial learning rate selection is still very important and heavily influences results. We have amended our supplementary material to demonstrate this (see Figure 10 in Appendix C). Additionally, as is discussed in (http://www.deeplearningbook.org, https://jmlr.csail.mit.edu/papers/volume13/bergstra12a/bergstra12a.pdf, and https://arxiv.org/pdf/2003.05689.pdf), the initial learning rate is a critical hyper-parameter (HP) to optimize and if there is one HP you should focus on, it is that. Thirdly, we acknowledge that scheduled learning rates for adaptive optimizers are popular, as the reviewer identifies, however our objective in this work is not to achieve superior performance and therefore scheduled learning rates were ignored. Future work would include multi-parameter optimization in which scheduled learning rates could be included.
>
> ### “In addition, as the learning rate...coarsely find a good learning rate and then refine”
>
> Although the reviewer is correct that grid searching can be used to optimize the initial learning rate, such a technique requires extensive domain knowledge, time, and is heavily influenced by the grid space considered (we refer to these papers listed from Reviewer #4: https://arxiv.org/abs/2007.01547, https://arxiv.org/abs/1910.11758, https://arxiv.org/abs/1910.05446). This highlights why our method is novel and important, as it requires no domain knowledge, human intuition, or manual intervention to set an initial grid space as it will dynamically expand or contract its search space based on our knowledge-gain-based metric. Further, when one considers the difference in time between our method and a grid searching method, if we consider a grid space of 20 learning rates such as our algorithm is initialized to, then a traditional grid search would consume 20 * 250 epochs = 5000 epochs, versus our method (which trials on average 30 epochs, see Figure 3) 30 * 5 = 150 epochs. In low resource or even single-gpu environments like the common practitioner, this reduction in total time is important. Early stopping could be used in traditional grid searching however generally, this would require a priori knowledge, whereas our method would not.
>
> ### “presented work would be more interesting...weight decay and convolution filter size and channels”
>
> We agree with the reviewer’s note here and identify that this includes our next step in future work. We note however that for the moment, our objective is to propose this work and new response surface (that any HP can be optimized against) as a proof of concept (which is also why we only considered the initial learning rate). Laying this foundation will be important for future work in this regard.
>
> ### “are all the convolution layers in a cnn used in equation 4? The definition of mode-3 and mode-4 is not clear to me”
>
> We note that our draft has been updated (with red text) to highlight changes made to clarify the reviewer’s comments. Yes, all the convolutional layers are used in equation 4, which is highlighted by the equation immediately prior to equation 4, where we sum over the layers in the network. Mode-3 and mode-4 simply refer to the input (3rd) and output (4th) dimension of the 4-D weight matrix for each layer. The mode-3 unfolded tensor is therefore simply the 4-D weight matrix, unfolded (reshaped) along the input dimension (3rd dimension), which is then analyzed as discussed in subsection 2.1.
>
> ** Note also that our anonymized code has been uploaded if the reviewers wish to examine it.

---

### Official Review · AnonReviewer2 · 2020-11-02
**Interesting paper, but not well written**

**Rating:** 5
**Confidence:** 4

**Review:**

** Summary

The paper proposes an efficient framework to search for the optimal initial learning rate to train neural networks. The key idea is to introduce Knowledge Gain, a metric derived from the singular values of each layer, to indicate the convergency quality of training. Taking advantage of the metric, a logarithmic grid search algorithm (AutoHyper) is proposed to search for the optimal learning rate according to Eq 5 via short-time training (e.g. for 5 epochs), which is demonstrated to be very efficient and take effect to some extent.


** Pros

1)	The paper studies a very important problem in neural network HPO: how to efficiently model and approximate the response function. The usage of Knowledge Gains as the proxy metric looks interesting and somewhat intuitive (however, I still have some concerns, see next).
2)	I appreciate the authors present a lot of empirical results, on different models, datasets and optimizers. Detailed experimental settings and numerical data are also provided, which could be convenient to reproduce.

** Cons

1)	The writing and presentation of the paper is not good.
2)	From Table 5 to Table 8, it seems the automatically searched results do not outperform the suggested baselines significantly in many cases.

** Technical concerns

In the definition of Knowledge Gain (Eq 3), though the distribution of singular values can be used as a metric to indicate the redundancy of a layer, to my knowledge it should be sensitive to the way of initialization; for example, an identity/orthogonal initialization may correspond to larger KG but random gaussian may result in smaller values. Furthermore, the evolving of KG during training may also vary from different layers, while in the formulation above Eq 4 it is just derived by averaging over all layers. So, I doubt whether the proposed metric is universally applicable for different situations. From the experiments, it seems the gain is not that significant.

** Presentation

Another of my major concerns is that the paper is not well written. I can hardly understand what exactly the paper suggests, since many formulations and figures are not clear, for example:

1)	In Eq 2, what is the exact definition of \hat{W}_d? What does “low-rank factorized matrix” refer to? It seems the formulation here is slightly different from those in Hosseini & Plataniotis (2020).
2)	At the beginning of Sec 2.2, when \bar{\mathcal{G}}_\lambda equals to 0? (from Eq 3, it seems never to be zero).
3)	In Eq 5, why to minimize (1 - \mathcal{Z})? Doesn’t smaller \mathcal{Z} correspond to better results? What is the meaning of the constraint in Eq 5?
4)	In Fig 1(b), how to calculate the histogram of \mathcal{Z} exactly?


================

Thanks for the rebuttal. I appreciate the authors' efforts on polishing the presentation. It clears some of my concerns. So, I raise my rating from 4 to 5. However, after reading the rebuttal and other reviewers' comments, I still feel that the contribution of the paper is not that significant,  since the search space just includes a scalar and the empirical improvements are not strong (Table 5 to Table 8). I think the authors may consider further applying their method to multi-dimension HPO problems (e.g. joint searching initial learning rate, momentum and weight decay) to verity the effectiveness on more challenging configurations.

---

> ### Author Response · Authors · 2020-11-14
> **Author Response**
>
> Thank you for taking the time and effort to review our paper and we appreciate your extensive comments and insight. To address the reviewer’s points...
>
> ** Note draft amendments are highlighted in red.
>
> ### “The writing and presentation of the paper is not good”
>
> Efforts have been made to improve in this regard, and we apologize, acknowledging this issue. The paper will be iteratively updated (the first iteration has been uploaded), with a final polish before November 24. We continue to welcome comments from the reviewer in augmenting our paper and making it more understandable.
>
> ### “From table 5 to table 8...baselines significantly in many cases”
>
> The reviewer is correct in this assessment and we appreciate their note, however we highlight that the intended objective of this work in not to significantly outperform baselines, but to introduce a novel methodology to tune the initial learning rate that requires no manual intervention or human intuition (for example in setting of the initial grid space and iteratively refining), and still achieve competitive results. We note that our method accomplishes this and except for a few cases, is always within the standard deviation of error in cases when it does not outperform the baseline. Further, our method comes at a significant reduction in cost, automatically tuning the initial learning rate in minutes or hours with no oversight, in contrast to say grid searching which can take days and extensive manual iteration. The baselines we test against use initial learning rates decided on by original authors through extensive HPO, which required significantly more time (epochs) than our method.
>
> ### In the definition of knowledge gain … from the experiments, it seems the gain is not that significant” + “in eq 2, what is the exact...formulation is slightly different..” + “at the beginning of sec 2.2...it seems never to be zero”
>
> The reviewer is correct in their assessment here, and we notice that the lack of explanation on our part has caused this confusion. Note that the draft has been updated to reflect the following comments. The distribution of singular values would be sensitive to the way of initialization if we were performing our decomposition (SVD) on only the unfolded tensor W_d, however we importantly factorize W_d into \hat{W}_d + E, where E is some perturbing noise that we subsequently ignore and \hat{W}_d is a low-rank factorized matrix of W_d. In this formulation, the randomness of initialization is captured by E and thus filtered when we ignore it, and we apply SVD on the low-rank factorized matrix \hat{W}_d. This formulation also permits \bar{\mathcal{G}}_{\lambda} to equal 0, as in the initial stages of training, due to the random initialization, \hat{W}_d will essentially be empty with all randomness contained within the noise E.
>
> Regarding the concern on averaging across layers, the reviewer is correct that KG varies across layers, however we wish to define a response model that is holistic to the network, and not locally optimal, which is why we take the average. Additionally, in our experimentation, we consider per-parameter adaptation such as AdaM, and therefore wish to select an initial learning rate that is globally optimal, and not per-layer optimal, to be more fair.
>
> Finally, \bar{\mathcal{G}}_{\lambda} will remain 0 if the learning rate is sufficiently small as still the weight matrix will be random as the gradients are not propagated enough, preventing beneficial updates. Only when the learning rate is large will the gradients be propagated to update our weight matrix.
>
> ### In Eq 5, why to ...meaning of the constraint in Eq5?”
>
> We apologize for the confusing notation and updated our draft in an attempt to resolve this confusion. We will continue to iterate on this formulation to ensure it is as clear as possible. In short, if one looks at Figure 1(a) or Figure 4 (the orange lines), we state that optimal learning rates are those that lie at the inception of the plateau-ing regions visualized in those graphs, and Eq 5 models this. It seems counterintuitive to minimize (1 - $\mathcal{Z}$), the reviewer is correct. Intuitively, our gradient constraint in Eq5 promotes learning rates  that lie along the plateau in the orange line of Figure 1(a) and Figure 4, while (1 - $\mathcal{Z}$), promotes smaller learning rates that lie towards the left of the graph, or at the inception of the plateau-ing region. In effect, we define Eq 5 such that its solution lies at the inception of this plateau-ing region.
>
> ### How to calculate the histogram?
>
> For every experimental setup (optimizer + dataset + network), there is a searching phase of a certain number of trialed learning rates (Figure 3).The histogram is taken by accumulating all the $\mathcal{Z}$ values for every trialed learning rate for every experimental setup.
>
> ** Note also that our anonymized code has been uploaded if the reviewers wish to examine it.

---

### Author Response · Authors · 2020-11-18
**General Comments to All Reviewers**

We thank all the reviewers for their valuable comments. We agree with the common sentiment that our paper is not well written and hard to understand (leading to some confusion) and have uploaded a revision as noted in our comment responses to address this. We would like to point out the following making notes on our revision as well in an attempt to resolve some confusion (note all revisions to our paper, including miscellaneous typos, are highlighted in red).
1. We would like to emphasize that our goal in this work is not to provide a method for HPO for superior performance, but rather provide a fully automatic tracking algorithm that can achieve competitive performance. It was a common note that the choice of searching bounds for learning rate (or any HP really) heavily influence final performance, which we agree with. This is exactly the problem that our method solves: it can automatically find a very competitive learning rate wihtout need for a priori knowledge, manual refinement or other. Subsequently, one could perform more intense HPO (random search, bayesian optimization, etc.), using the autoHyper suggested learning rate as a search range, if truly superior performance is required. Our method therefore saves practitioners large amounts of time and removes the need for iterative refinement of optimal search spaces.
2. In regards to some confusion around our metric development, we have made revision to the theoretical work in section 2.
3. Our code has been uploaded for those interested: Please see the `src/autohyper` folder
4. We are currently conducting additional baselines comparing our method to grid searching, as we do agree with the reviewers that these additional results will prove beneficial. We will comment on those results when they are completed.
5. Figures 5 and 10 (now Figure 11 in the draft) have been updated to be more clear.

Thank you

---

### Author Response · Authors · 2020-11-24
**Comment to All Reviewers:  Random Search Comparison**

In addition to the previous general comment, we would like to point out that the inclusion of our SOTA comparison (Random Search), has now been included in the Appendix (Appendix D). Once again, all changes are highlighted in red. This experimental addition compares our method (autoHyper) against random search. Details of the experiment are discussed in the appendix. Note that we will be moving the results to the main draft before the deadline. In a highlighted note, autoHyper proves superior to random search when random search is given the same epoch budget and trial count.

Thank you and please ask us any questions still outstanding.

---

### Author Response · Authors · 2020-11-25
**Final Revision**

We would like to notify the reviewers that the final revision of our paper has now been uploaded. Aside from minor fixes, the three major changes are:
1. Inclusion of random search comparison in main draft (see experiments)
2. amendment to our metric formulation (section 2.1 & 2.2, pages 3/4). Specifically, we note that our metric is in fact a summation of the zero-valued knowledge gains as we write in the paper, and our formulation before was incorrect/confusing. We hope this updated formulation is more clear.
3. Amendment to Appendix D regarding the random search comparison. Included some additional discussion notes there as well as results.

---

### Decision · Program_Chairs · 2021-01-07
**Final Decision**

**Decision:**

Reject

**Comment:**

This paper studies the important problem of efficiently identifying good hyperparameters for convolutional neural networks. The proposed approach is based on using an SVD of unfolded weight tensors to build a response surface that can be optimized with a dynamic tracking algorithm. Reviewers raised a number of concerns which were not fully addressed in rebuttals and lead me to recommend rejecting this work. In particular: focus on single hyperparameter (learning rate) made it unclear whether the proposed approach could actually be used for other hyperparameters or to jointly optimize combination of hyperparameters, empirical improvements even for learning rate are not strong and baselines are weak, and concern that initial success early in training (5 epochs) may not lead to generalization late in training. Additionally, there were several concerns around the clarity of the presentation, which I also found hard to follow: how is KG related to information-theoretic metrics, why is the particular form of averaging across layers reasonable, and how is it related to other generalization / performance metrics? With additional experiments on other hyperparameters (for example L2 regularization), I think the work would be greatly strengthened.